# Zero-sum beliefs and the avoidance of political conversations
F. Katelynn Boland ⓘ & Shai Davidai ⓘ ✉

Although researchers have argued that exposure to diverse views may help reduce political divisions in society, people often avoid discussing politics with ideologically opposed others. We investigate the avoidance of political conversations surrounding highly contested elections in Israel and the U.S. Specifically, we examine the relationship between people's belief that politics is a zero-sum game and their tendency to avoid talking about politics with ideologically opposed others. In two studies conducted in the days leading up to their countries' elections, we found that Israeli and American voters who view politics as zero-sum avoided political discussions with ideologically opposed others. Furthermore, zero-sum beliefs about politics statistically predicted the avoidance of political conversations through two distinct mechanisms: perceived conflict and a lack of receptiveness to opposing views. Finally, in a longitudinal design, we found that zero-sum beliefs about politics statistically and robustly predicted the avoidance of political conversation one week later.

On November 10, 2022, as Ann Coulter (a controversial conservative pundit) was preparing to host a Q&A about the U.S. midterm elections at Cornell University, she was met with vocal student opposition that ultimately forced the organizers to cancel her talk[1]. Although such protests often garner significant public attention[2], liberal students are not unique in their strident opposition to conservative speakers. In fact, disruptions of campus events featuring politically divisive figures are not limited to any political orientation[3], and controversial liberal speakers have been similarly forced to cancel their campus appearances in the face of vocal conservative opposition[4]. With both liberal and conservative students refusing to engage with ideologically opposed others, the deep-seated polarization of U.S. politics and the escalating tensions surrounding political discourse on U.S. campuses seem to have reached new heights.

Of course, the active avoidance of political discourse with ideologically opposed others is not unique to college campuses nor to the U.S. According to the Pew Research Center, 45% of Americans report that they have actively avoided talking about politics with someone because of their ideology[5], and an analysis of over 93,000 people across 64 countries found that about one out of every three respondents *never* discuss politics with their friends (Supplementary Note 1)[6]. Many people, it seems, avoid talking politics whenever they have the chance to do so[7].

The fact that people avoid talking about politics with ideologically opposed others is consequential. Engaging in political conversations and exposing oneself to opposing views can reduce affective polarization[8], increase deliberation, tolerance, and appreciation of diverse viewpoints[9–11],

broaden political knowledge and awareness[12–14], and increase people's willingness to update their beliefs when facing new information[15]. Yet, due to rising animosity among political partisans[16–21], people often avoid talking about politics[22]. Consequently, this avoidance creates echo chambers that deny and discredit opposing views[23] and thus perpetuates political division in society.

Why do people avoid political conversations? In this work, we examine the avoidance of political conversations in the days and weeks leading up to two highly contested elections in Israel and the U.S.—a time when politics, politicians, and political conversations are clearly top-of-mind, highly salient, and of extreme importance. Specifically, we examine whether seeing politics as zero-sum statistically predicts people's tendency to avoid talking about it with ideologically opposed others. While some issues may, in fact, be zero-sum (e.g., a party that refuses to recognize its counterpart's existence or legitimacy), we argue that that the general tendency to view politics as zero-sum statistically reduces people's willingness to talk about it.

There are various potential reasons for why people may actively avoid engaging in political conversations. For instance, individual differences in the aversion to conflict, people's desire to maintain harmony in their relationships, and individual differences in susceptibility to cognitive phenomena such as the confirmation bias may all lead people to avoid potentially divisive political discussions[24–26]. Similarly, contextual factors such as the overall political climate and the perceived intensity of partisan divisions in society may also shape people's avoidance of political conversations[27–29]. Indeed, as political debates become increasingly

Columbia Business School, New York, NY, USA. ✉e-mail: sd3311@columbia.edu

polarized and hostile, people may opt to retreat from engaging in such conversations in order to sidestep potential conflicts and the personal psychological toll that doing so entails[30]. More broadly, sociocultural influences such as one's social networks and community dynamics may similarly play a part in the avoidance of political debates, fostering fear of judgment and alienation from one's peers and colleagues.

While previous research has focused on various dispositional and situational factors that help explain people's avoidance of political conversations, we propose that specific beliefs about the nature of politics may similarly shape such avoidance. Specifically, we suggest that zero-sum beliefs about politics—the implicit or explicit belief that "any gain made by one party must result in an equivalent loss for another party"[31]—may contribute to people's avoidance of political conversations. Although some researchers have treated these beliefs as a general mindset about the world[32–36], others have examined them in more narrowly defined contexts such as immigration[37,38], race and gender[39–42], corporate profits[43] and international trade[44]. Building on this domain-specific approach to zero-sum beliefs, we examine whether viewing politics as zero-sum predicts the avoidance of political conversations. Since adversarial conversations can strain relationships[45], and given that the belief that zero-sum situations lead to conflict[46], we explore whether people who view politics as zero-sum actively avoid talking about it with ideologically opposed others (while studying the avoidance of politically charged topics is equally interesting, in this paper we focus on studying the avoidance of politically opposed others).

To be clear, we conceptualize the belief that politics is a zero-sum game as a malleable and context dependent mindset that is affected by the environment in which people find themselves[31]. Since perceptions of inter-partisan relations shift as a result of increasing political polarization, the propensity to view politics as zero-sum may similarly change, shifting people's focus from compromise and cooperation to heightened competition and antagonism. Thus, understanding people's zero-sum beliefs about politics may help tackle broader issues relating to the context in which they evolve.

There are two main reasons why zero-sum beliefs may predict the avoidance of political conversations. First, people who see politics as zero-sum may worry that talking about it creates conflict and hostility. Since zero-sum beliefs make animosity seem unavoidable, widespread, and normative[47,48], people who hold such beliefs may view political conversations as inevitably conflict prone and something that cannot be easily avoided. Indeed, since partisans overestimate their political outgroup's hostility[49,50] and view political conversations as inherently confrontational[51], they may worry that talking about politics necessarily harms relationships[52]. Of course, viewing politics as zero-sum is inherently different from merely seeing it as a two-sided conflict[53,54]. Whereas many two-sided conflicts have potential (albeit non-obvious) 'win-win' solutions[55,56], zero-sum beliefs about politics specifically relate to a view of the political sphere as a zero-sum conflict in which people can only gain at others' expense. Thus, to the extent that people view politics as specifically a zero-sum conflict, they may be more likely to disengage from talking about it, viewing any potential conversation as leading to interpersonal animosity. Consequently, the tendency to withdraw from conflict[57] may encourage those who see politics as zero-sum to avoid talking about it. Just as the fear of conflict explains why zero-sum beliefs lead people to avoid negotiations[58], we hypothesized that concerns about the eruption of potential conflict and hostility would explain the effect of zero-sum beliefs about politics on the avoidance of political conversations.

Second, the avoidance of political conversations may also be affected by people's receptiveness to opposing views (i.e., the motivational tendency "to access, consider, and evaluate contradictory opinions in a relatively impartial manner"). Specifically, we argue that people who see politics as zero-sum may be especially unreceptive to opposing views and thus avoid talking with ideologically opposed others. Low receptiveness to opposing views predicts resistance to counter-attitudinal information[59,60] and a rejection of those who do not share one's beliefs[61]. For instance, a study examining how Wikipedia editors resolve their differences found that those who were low in conversational receptiveness were more prone to attack and be attacked by others on the site[62]. In contrast, multicultural experiences and feeling close to one's conversation partner increase receptiveness and openness to opposing views[63–65]. Thus, in the same way that zero-sum beliefs are associated with close-mindedness toward immigration, LGBTQ rights, and gender and racial equality[37,40,66,67], seeing politics as zero-sum may predict a lower receptiveness to opposing views. Consequently, this lack of receptiveness may help explain the relationship between the tendency to view politics as zero-sum and people's avoidance of talking about politics with ideologically opposed others.

In sum, we hypothesize that zero-sum beliefs about politics predict people's tendency to avoid political conversations with ideologically opposed others. We argue that people who see politics as zero-sum tend to exhibit reduced receptivity to opposing views and, independently, anticipate hostile conflict within political conversations, which may contribute to avoidance of such discussions. Thus, by investigating the correlates and potential antecedents of people's avoidance of political conversations, we hope to build a much-needed bridge between the literature on zero-sum beliefs and the literature on political psychology.

## Methods
Although not pre-registered, analyses were conducted after data collection was complete, and all measures and conditions are reported. The sample sizes for both studies were determined in advance (while the sample for Study 1 was based on budgetary constraints, the sample for Study 2 was based on the observed effect sizes in Study 1). All relevant ethical regulations were followed and informed consent was obtained in accordance with the Columbia University Institutional Review Board.

### Study 1
**Participants.** Four hundred and three Israeli residents (recruited via the Midgam Project Web Panel) completed the study on the day before Israel's 26th election for parliament (October 31, 2022; $M_{age} = 40.59$, $SD = 11.76$; 51.7% women, 48.3% men; 98.5% Jewish, 1.5% other). A post-hoc sensitivity power analyses revealed that this sample size allows 80% power to detect an effect size for a regression as small as $f^2 = 0.024$.

**Procedure.** Participants were first asked to indicate, on a categorical variable, their political party affiliation, identifying the political party that they most closely support among the 13 different parties that participated in the 2022 elections to Israel's parliament (the "Knesset"; "Generally speaking, which of the following political parties do you support?": Ha'Likkud, Ha'Avoda, Yesh Atid, HaZionot Ha'Datit, Meretz, Ra'am, Ha'Machane Ha'Mamlachti, Hadash-Ta'al, Shas, Ha'Bait Ha'Yehudi, Yahadut Ha'Torah, Balad, Israel Beitenu, and Other). They next completed, in random order, a six-item measure of zero-sum beliefs about politics (e.g., "When one political party gains it inevitably comes at another party's expense"; 1-Strongly disagree, 7-Strongly agree; α = 0.69), a Hebrew translation of the 18-item Receptiveness to Opposing Views scale (α = 0.82)[59], a five-item measure of perceived conflict (e.g., "Talking about politics always creates harmful conflict between relatives"; 1-Strongly disagree, 7-Strongly agree; α = 0.94)[39] and a four-item measure of their avoidance of political conversations with ideologically opposed others *in the past month* (e.g., "In the past month, I avoided talking about politics with family members with whom I disagree"; 1-Strongly disagree, 7-Strongly agree; α = 0.91) (see Supplementary Materials). Finally, participants reported their age, gender, religion, income, and whether they intend to vote in the upcoming elections (*yes, no,* or *I have yet to decide*). As reported below, we analyzed the data using a series of linear regression analyses examining the zero-order correlations between zero-sum beliefs and avoidance of political conversation, the effect of zero-sum beliefs on avoidance of political conversation when controlling for political party affiliation and its interaction with zero-sum beliefs, and a bootstrapped mediation analysis predicting conversation avoidance from zero-sum beliefs and the two potential mediators (perceived conflict and receptiveness to opposing views).

## Study 2

**Participants.** Five hundred ninety-eight U.S. residents (recruited via Prolific Academic) completed the first wave of the study on October 31, 2022, 1 week before the 2022 U.S. Midterm Elections. We excluded 19 participants who failed an attention check (participants were asked to write the number of letters in the word "computer"), resulting in a sample of 579 participants ($M_{age}$ = 40.99, $SD$ = 14.61; 51.1% women, 47.3% men, 1.5% other; 78.6% White, 6.0% Asian/Asian-American, 5.1% Latino/Hispanic American, 3.9% Black/African American, <1% Native Hawaiian/Pacific Islander, 6.2% other). This sample size allows 80% power to detect an effect size as small as $f^2$ = 0.017. Of these, 480 participants completed the second wave of the survey a week later, on the day before the midterm elections (November 7, 2022) ($M_{age}$ = 41.83, $SD$ = 14.37; 49.3% women, 49.3% men, 1.5% other; 78.3% White, 6.3% Asian/Asian-American, 5.4% Latino/Hispanic American, 4.0% Black/African American, <1% Native Hawaiian/Pacific Islander, 5.8% other). A post-hoc sensitivity power analyses revealed that this sample size allows 80% power to detect an effect size for a regression as small as $f^2$ = 0.020.

**Procedure (Time 1).** Participants indicated, on a three-level categorical variable, their political party affiliation by reporting the political party with which they most closely identify in the U.S. political map ("Generally speaking, how do you usually think of yourself in terms of political affiliation?"; Republican, Democrat, or Independent). In addition, participants indicated the strength of their political affiliation (for Republicans and Democrats; "Would you call yourself a strong Republican/Democrat, or not a very strong Republican/Democrat?") or their general leanings (for independents; "If you had to choose, would you say that you lean more towards Republicans or Democrats?"). Below, we report the results using the three-level categorical measure, although the results remain virtually unchanged when using the continuous measure of participants' strength of political party affiliation.

Next, participants completed, in random order, the same measures from Study 1: a six-item measure of zero-sum beliefs about politics ($\alpha$ = 0.86), the 18-item Receptiveness to Opposing Views scale ($\alpha$ = 0.90)[59], a five-item measure of perceived conflict in political conversations ($\alpha$ = 0.94), and a four-item measure of their avoidance of political conversations with ideologically opposed in the past month ($\alpha$ = 0.87). Following, participants completed, in counterbalanced order, the ten-item Big Five Inventory scale ($\alpha$ = 0.70)[68] and the seven-item Perspective-Taking scale ($\alpha$ = 0.85)[69,70]- a psychological construct that has been shown to strongly correlate with people's receptiveness to opposing views and their prejudice towards outgroup members. In addition, participants completed an attention check (see above), and self-reported their age, gender, race, income, political ideology (on a 7-point Likert scale, ranging from 1-Very liberal to 7-Very conservative, with 4-Neither liberal/conservative as the midpoint), and whether they plan to vote in the upcoming 2022 midterm elections (*yes, no,* or *haven't decided*).

**Procedure (Time 2).** Participants completed, in counterbalanced order, the same independent and dependent variables from Time 1: the six-item measure of zero-sum beliefs about politics ($\alpha$ = 0.89) and the four-item measure of avoidance of political conversations ($\alpha$ = 0.90). Unlike Time 1, participants reported how much they avoided talking about politics with ideologically opposed others *in the preceding week* since the first wave of the survey (e.g., "In the past week I have avoided talking politics with family members with whom I disagree").

As reported below, for each separate time point, we analyzed the data using a series of linear regression analyses examining the zero-order correlations between zero-sum beliefs and avoidance of political conversation, the effect of zero-sum beliefs on avoidance of political conversation when controlling for political party affiliation, political ideology, personality traits, and self-reported perspective taking abilities, and a bootstrapped mediation analysis predicting conversation avoidance from zero-sum beliefs and the two potential mediators (perceived conflict and receptiveness to opposing

views). In addition, we conducted a time-lagged correlation analysis to examine the predictive power of zero-sum beliefs at Time 1 on avoidance of political conversation at Time 2.

## Reporting summary

Further information on research design is available in the Nature Portfolio Reporting Summary linked to this article.

## Results

We conducted Study 1 on October 31, 2022—one day before Israel's 26th general elections for parliament (the "Knesset"). As noted above, we asked a sample of 403 eligible Israeli voters to report, in random order, their zero-sum beliefs about politics, whether they avoided discussing politics in the past month with ideologically opposed others, their general receptiveness to opposing views[59], and the extent to which they saw political conversations as conflict-prone. To begin, we examined whether zero-sum beliefs about politics statistically predicted Israeli voters' avoidance of political conversations in the month leading up to the elections. A linear regression controlling for participants' political party affiliation (i.e., which of the 13 parties running in the coming elections they supported) revealed a significant effect of zero-sum beliefs on the avoidance of political conversations ($b$ = 0.23, 95% CI[0.01, 0.44], $t(390)$ = 2.07, $p$ = 0.039; Table 1, Model 1), although the zero-order correlation was only marginally significant ($b$ = 0.21, 95% CI[−0.002, 0.42], $t(403)$ = 1.95, $p$ = 0.052). Importantly, a multiple regression analysis predicting conversation avoidance from zero-sum beliefs, political party affiliation, and their interaction did not find evidence of moderation by political party affiliation (Table 1, Model 2).

Next, we examined the two potential mediators for the relationship between zero-sum beliefs and the avoidance of political conversations. As hypothesized, zero-sum beliefs statistically predicted how much conflict participants' expected to experience in political conversations ($b$ = 0.52, 95% CI[0.35, 0.69], $t(401)$ = 6.13, $p$ < 0.001), and these perceptions of conflict mediated the relationship between zero-sum beliefs and the avoidance of political conversations (indirect effect: $b$ = 0.29, 95% CI[0.18, 0.42], $p$ < 0.001; direct effect: $b$ = −0.09, 95% CI[−0.29, 0.10], $p$ = 0.348). Similarly, zero-sum beliefs were negatively associated with receptiveness to opposing views ($b$ = −0.17, 95% CI[−0.23, −0.11], $t(388)$ = −5.32, $p$ < 0.001) which mediated the effect of such beliefs on participants' avoidance of political conversations (indirect effect: $b$ = 0.14, 95% CI[0.07, 0.23], $p$ < 0.001; direct effect: $b$ = 0.11, 95% CI[−0.12, 0.35], $p$ = 0.344).

Finally, a multiple mediation analysis examined each mediator's unique contribution to the relationship between zero-sum beliefs and the avoidance of political conversation. As shown in Fig. 1, this analysis revealed a significant indirect effect through perceived conflict ($b$ = 0.27, 95% CI[0.16, 0.37], $p$ < 0.001) and a much smaller indirect effect through low receptiveness ($b$ = 0.07, 95% CI[0.02, 0.13], $p$ = 0.013). Specifically, we found that zero-sum beliefs predicted participants' perceived conflict ($b$ = 0.52, $p$ < 0.001) and receptiveness to opposing views ($b$ = −0.17, $p$ < 0.001) which, in turn, predicted their avoidance of political conversations ($b$ = 0.52, $p$ < 0.001, and $b$ = −0.45, $p$ < 0.005, respectively). In contrast, the direct effect of zero-sum beliefs on the avoidance of political conversations was not significant ($b$ = −0.13, 95% CI[−0.33, 0.06], $p$ = 0.183). Thus, participants' tendency to view politics as a zero-sum game statistically predicted their expectations of conflict in political conversations and (to a less extent) their reduced receptiveness to opposing views. Consequently, perceptions of conflict and a low receptiveness to others' views explain the relationship between zero-sum beliefs and participants' avoidance of political conversations.

The more Israeli participants viewed politics as zero-sum, the more they avoided talking about it with ideologically opposed others during the month leading up to the Israeli elections. Study 2 replicated and extended these findings with a sample of 579 U.S. participants who completed, 1 week before the 2022 U.S. midterm elections, the first wave of our study (Time 1). As before, a linear regression analysis predicting conversation avoidance found that zero-sum beliefs statistically predicted the avoidance of political

**Table 1 | Study 1: Linear regression predicting the avoidance of political conversations from zero-sum beliefs and political party affiliation (Model 1) as well as their interaction (Model 2)**

|  | Predictor | b | p value | b 95% CI [LL, UL] | sr² | sr² 95% CI [LL, UL] | Fit |
|---|---|---|---|---|---|---|---|
| Model 1 | Intercept | 2.73 | 7.5e-08 | [1.75, 3.71] |  |  |  |
|  | Zero-Sum Beliefs | 0.23 | 0.0391 | [0.01, 0.44] | 0.01 | [−0.01, 0.03] |  |
|  | Party 2 | 0.00 | 0.9934 | [−0.53, 0.54] | 0.00 | [−0.00, 0.00] |  |
|  | Party 3 | −0.27 | 0.4095 | [−0.90, 0.37] | 0.00 | [−0.01, 0.01] |  |
|  | Party 4 | 0.03 | 0.9328 | [−0.71, 0.77] | 0.00 | [−0.00, 0.00] |  |
|  | Party 5 | −0.32 | 0.4535 | [−1.15, 0.51] | 0.00 | [−0.01, 0.01] |  |
|  | Party 6 | −0.08 | 0.8649 | [−0.94, 0.79] | 0.00 | [−0.00, 0.00] |  |
|  | Party 7 | 0.13 | 0.7753 | [−0.79, 1.05] | 0.00 | [−0.00, 0.00] |  |
|  | Party 8 | 0.48 | 0.2895 | [−0.41, 1.38] | 0.00 | [−0.01, 0.01] |  |
|  | Party 9 | 0.80 | 0.1019 | [−0.16, 1.76] | 0.01 | [−0.01, 0.02] |  |
|  | Party 10 | 0.44 | 0.8165 | [−3.25, 4.13] | 0.00 | [−0.00, 0.00] |  |
|  | Party 11 | −1.10 | 0.5574 | [−4.79, 2.59] | 0.00 | [−0.00, 0.01] |  |
|  | Party 12 | 0.06 | 0.9220 | [−1.16, 1.28] | 0.00 | [−0.00, 0.00] |  |
|  | Party 14 | 0.01 | 0.9868 | [−0.95, 0.97] | 0.00 | [−0.00, 0.00] |  |
|  |  |  |  |  |  |  | $R^2 = 0.02695\%$ CI[0.00, 0.03] |
| Model 2 | Intercept | 2.45 | 0.0050 | [0.74, 4.15] |  |  |  |
|  | Zero-Sum Beliefs | 0.29 | 0.1419 | [−0.10, 0.69] | 0.01 | [−0.01, 0.02] |  |
|  | Party2 | 0.29 | 0.8410 | [−2.51, 3.08] | 0.00 | [−0.00, 0.00] |  |
|  | Party3 | 1.12 | 0.4392 | [−1.73, 3.98] | 0.00 | [−0.01, 0.01] |  |
|  | Party4 | −3.16 | 0.0947 | [−6.88, 0.55] | 0.01 | [−0.01, 0.02] |  |
|  | Party5 | −2.04 | 0.2922 | [−5.84, 1.76] | 0.00 | [−0.01, 0.01] |  |
|  | Party6 | 0.70 | 0.7814 | [−4.28, 5.69] | 0.00 | [−0.00, 0.00] |  |
|  | Party7 | 0.27 | 0.8961 | [−3.76, 4.30] | 0.00 | [−0.00, 0.00] |  |
|  | Party8 | 1.96 | 0.4136 | [−2.75, 6.67] | 0.00 | [−0.01, 0.01] |  |
|  | Party9 | 5.13 | 0.0764 | [−0.55, 10.80] | 0.01 | [−0.01, 0.02] |  |
|  | Party10 | 0.47 | 0.8005 | [−3.22, 4.17] | 0.00 | [−0.00, 0.00] |  |
|  | Party11 | −1.07 | 0.5673 | [−4.76, 2.61] | 0.00 | [−0.00, 0.01] |  |
|  | Party12 | 3.84 | 0.4707 | [−6.61, 14.29] | 0.00 | [−0.01, 0.01] |  |
|  | Party14 | 1.74 | 0.3364 | [−1.81, 5.28] | 0.00 | [−0.01, 0.01] |  |
|  | Zero-Sum Beliefs x Party2 | −0.07 | 0.8559 | [−0.74, 0.61] | 0.00 | [−0.00, 0.00] |  |
|  | Zero-Sum Beliefs x Party3 | −0.34 | 0.3239 | [−1.00, 0.33] | 0.00 | [−0.01, 0.01] |  |
|  | Zero-Sum Beliefs x Party4 | 0.80 | 0.0806 | [−0.10, 1.69] | 0.01 | [−0.01, 0.02] |  |
|  | Zero-Sum Beliefs x Party5 | 0.40 | 0.3633 | [−0.47, 1.27] | 0.00 | [−0.01, 0.01] |  |
|  | Zero-Sum Beliefs x Party6 | −0.20 | 0.7623 | [−1.49, 1.09] | 0.00 | [−0.00, 0.00] |  |
|  | Zero-Sum Beliefs x Party7 | −0.03 | 0.9582 | [−1.05, 0.99] | 0.00 | [−0.00, 0.00] |  |
|  | Zero-Sum Beliefs x Party8 | −0.37 | 0.5332 | [−1.55, 0.80] | 0.00 | [−0.00, 0.01] |  |
|  | Zero-Sum Beliefs x Party9 | −1.08 | 0.1287 | [−2.47, 0.31] | 0.01 | [−0.01, 0.02] |  |
|  | Zero-Sum Beliefs x Party10 | NA | NA | [NA, NA] | NA | [NA, NA] |  |
|  | Zero-Sum Beliefs x Party11 | NA | NA | [NA, NA] | NA | [NA, NA] |  |
|  | Zero-Sum Beliefs x Party12 | −1.02 | 0.4779 | [−3.84, 1.80] | 0.00 | [−0.01, 0.01] |  |
|  | Zero-Sum Beliefs x Party14 | −0.42 | 0.3193 | [−1.24, 0.40] | 0.00 | [−0.01, 0.01] |  |
|  |  |  |  |  |  |  | $R^2 = 0.053$ 95% CI[0.00, 0.04] |

The effect of zero-sum beliefs becomes marginal when interaction terms are included in the model.

conversations ($b = 0.36$, 95% CI[0.19, 0.52], $t(576) = 4.15$, $p < 0.001$) and that this was true even when controlling for whether participants' identified as Republicans, Democrats, or Independent ($b = 0.37$, 95% CI[0.21, 0.54], $t(574) = 4.38$, $p < 0.001$; Table 2, Model 1). Providing additional evidence of the robustness of this effect, an additional a multiple linear regression predicting the avoidance of political conversations while controlling for participants' personality traits, self-reported perspective-taking abilities, and a continuous measure of their political ideology (very liberal to very conservative) found a significant effect of positive zero-sum beliefs on conversational avoidance ($b = 0.33$, 95% CI[0.16, 0.50], $t(559) = 3.87$, $p < 0.001$) as well as a positive effect of trait-level neuroticism ($b = 0.21$, 95% CI[0.08, 0.33], $p < 0.001$) (Table 2, Model 2). Moreover, there was no statistically significant association between political extremism (operationalized as the absolute distance of participants' liberalism/conservatism from the

**Fig. 1 | The association of zero-sum beliefs and avoidance of policital conversation in Israel.** Legend: The role of perceived conflict and receptiveness to opposing views in the relationship between zero-sum beliefs and the avoidance of political conversation among Israeli voters (Study 1); *n* = 391.

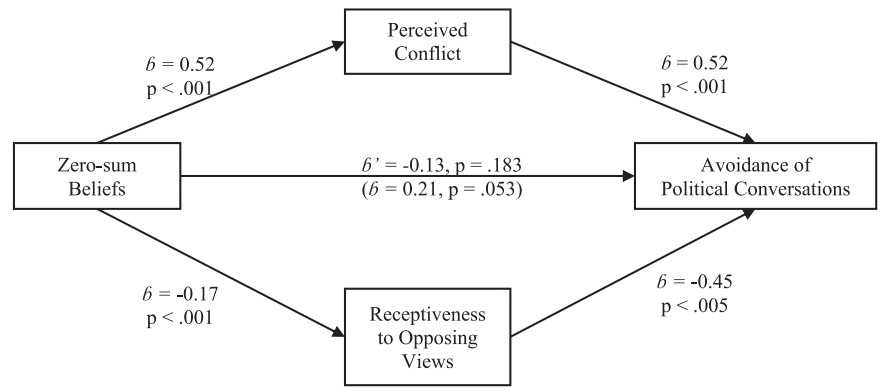

**Table 2 | Study 2, Time 1: Linear regressions predicting avoidance of political conversations from zero-sum beliefs and political party affiliation (three-level categorical variable: Democrat vs. Republican vs. Independent; Model 1), from zero-sum beliefs, Big-Five personality traits, self-reported perspective-taking abilities, and political orientation (7-point Likert scale: very liberal—very conservative; Model 2), and from zero-sum beliefs and political extremism (a continuous, 4-point scale denoting the absolute distance of their liberalism/conservatism from the midpoint of the political ideology scale; Model 3)**

| | Predictor | *b* | *p* value | 95% CI | *sr*² | *sr*² 95% CI | Fit |
|---|---|---|---|---|---|---|---|
| Model 1 | Intercept | 3.15 | 2.90e-16 | [2.41, 3.88] | | | |
| | Zero-Sum Beliefs | 0.37 | 1.39e-05 | [0.21, 0.54] | 0.03 | [0.00, 0.06] | |
| | Republican | −0.28 | 0.0281 | [−0.53, −0.03] | 0.01 | [−0.01, 0.02] | |
| | Independent/Other | 0.71 | 0.1117 | [−0.17, 1.59] | 0.00 | [−0.01, 0.01] | |
| | | | | | | | *R*² = 0.043 95% CI[0.01, 0.08] |
| Model 2 | Intercept | 2.84 | 5.6e-05 | [1.47, 4.22] | | | |
| | Zero-Sum Beliefs | 0.33 | 0.0001 | [0.16, 0.50] | 0.02 | [0.00, 0.05] | |
| | Political Orientation | −0.04 | 0.1827 | [−0.11, 0.02] | 0.00 | [−0.01, 0.01] | |
| | Extraversion | −0.05 | 0.4122 | [−0.16, 0.07] | 0.00 | [−0.00, 0.01] | |
| | Conscientiousness | 0.08 | 0.3326 | [−0.08, 0.24] | 0.00 | [−0.00, 0.01] | |
| | Agreeableness | −0.04 | 0.5441 | [−0.19, 0.10] | 0.00 | [−0.00, 0.00] | |
| | Neuroticism | 0.21 | 0.0011 | [0.08, 0.33] | 0.02 | [−0.00, 0.04] | |
| | Openness | 0.06 | 0.3586 | [−0.07, 0.20] | 0.00 | [−0.00, 0.01] | |
| | Perspective-Taking | −0.09 | 0.3559 | [−0.27, 0.10] | 0.00 | [−0.00, 0.01] | |
| | | | | | | | *R*² = 0.068 95% CI[0.02, 0.10] |
| Model 3 | Intercept | 2.99 | 6.63e-14 | [2.23, 3.76] | | | |
| | Zero-Sum Beliefs | 0.35 | 5.47e-05 | [0.18, 0.52] | 0.03 | [0.00, 0.05] | |
| | Political Extremism | 0.08 | 0.3150 | [−0.07, 0.22] | 0.00 | [−0.00, 0.01] | |
| | | | | | | | *R*² = 0.031 95% CI[.01, 0.06] |

midpoint of the political ideology scale) and participants' avoidance of political conversation ($b = 0.08$, 95% CI[−0.07, 0.22], $t(575) = 1.01$, $p = 0.315$ (Table 2, Model 3).

As before, we did not find evidence of moderation of the predictive effects of zero-sum beliefs by participants' political party affiliation (operationalized as a three-level categorical variable, based on their identification as Republicans, Democrats, or Independents; Table 3, Model 1) or their political orientation (operationalized as a continuous 7-point Likert scale, based on their level of political liberalism/conservatism; Table 3, Model 2).

We next examined the proposed mediators for the relationship between zero-sum beliefs and the avoidance of political conversations. As before, a linear regression found that zero-sum beliefs statistically predicted perceptions of conflict in political conversations ($b = 0.45$, 95% CI[0.31, 0.59], $t(577) = 6.43$, $p < 0.001$), which subsequently mediated the relationship between zero-sum beliefs and the avoidance of such conversations (indirect effect: $b = 0.28$, 95% CI[0.18, 0.38], $p < 0.001$; direct effect: $b = 0.06$, 95% CI[−0.11, 0.24], $p = 0.45$). Similarly, participants' zero-sum beliefs

were significantly associated with their receptiveness to opposing views ($b = −0.31$, 95% CI[−0.42, −0.21], $t(577) = −5.80$, $p < 0.001$), which mediated the effect on avoidance of political conversations (indirect effect: $b = 0.15$, 95% CI[0.08, 0.22], $p < 0.001$; direct effect: $b = 0.19$, 95% CI[0, 0.39], $p = 0.046$).

Finally, a multiple mediation analysis of each mediator's unique contribution provided strong evidence of simultaneous mediation. As shown in Fig. 2, we found a significant indirect effect through perceived conflict ($b = 0.25$, 95% CI[0.16, 0.34], $p < 0.001$), a smaller indirect effect through receptiveness to opposing views ($b = 0.07$, 95% CI[0.02, 0.11], $p = 0.003$), and an insignificant direct effect ($b = 0.02$, 95% CI[−0.13, 0.18], $p = 0.777$). Specifically, zero-sum beliefs predicted participants' perceived conflict ($b = 0.45$, $p < 0.001$) and receptiveness to opposing views ($b = −0.31$, $p < 0.001$) which, in turn, predicted their avoidance of political conversations ($b = 0.55$, $p < 0.001$, and $b = −0.22$, $p < 0.005$, respectively). Replicating Study 1, participants who saw politics as zero-sum tended to view political discussions as conflict-prone and exhibited reduced receptiveness to

**Table 3 | Study 1: Linear regressions predicting the avoidance of political conversations from zero-sum beliefs, political party affiliation, and their interaction (three-level categorical variable: Democrat vs. Republican vs. Independent; Model 1) and zero-sum beliefs, political orientation, and their interaction (7-point Likert scale: very liberal—very conservative; Model 2)**

| | Predictor | $b$ | $p$ value | $b$ 95% CI | $sr^2$ | $sr^2$ 95% CI | Fit |
|---|---|---|---|---|---|---|---|
| Model 1 | Intercept | 3.69 | 8.81e-11 | [2.59, 4.79] | | | |
| | Zero-Sum Beliefs | 0.25 | 0.0581 | [−0.01, 0.50] | 0.01 | [−0.01, 0.02] | |
| | Republican | −1.22 | 0.1090 | [−2.72, 0.27] | 0.00 | [−0.01, 0.01] | |
| | Independent/Other | −0.47 | 0.7894 | [−3.92, 2.98] | 0.00 | [−0.00, 0.00] | |
| | Zero-Sum Beliefs x Republican | 0.22 | 0.2103 | [−0.12, 0.57] | 0.00 | [−0.01, 0.01] | |
| | Zero-Sum Beliefs x Independent/Other | 0.29 | 0.4958 | [−0.54, 1.12] | 0.00 | [−0.00, 0.01] | |
| | | | | | | | $R^2 = 0.046$ 95% CI[0.01, 0.08] |
| Model 2 | Intercept | 4.35 | 5.14e-08 | [2.80, 5.89] | | | |
| | Zero-Sum Beliefs | 0.13 | 0.4754 | [−0.23, 0.49] | 0.00 | [−0.00, 0.01] | |
| | Political Orientation | −0.33 | 0.0574 | [−0.68, 0.01] | 0.01 | [−0.01, 0.02] | |
| | Zero-Sum Beliefs x Political Orientation | 0.06 | 0.1292 | [−0.02, 0.14] | 0.00 | [−0.01, 0.01] | |
| | | | | | | | $R^2 = 0.042$ 95% CI[0.01, 0.07] |

The main effect of zero-sum beliefs becomes non-significant when all interaction terms are included in the model.

**Fig. 2 | The association of zero-sum beliefs and avoidance of policital conversation in the U.S.**
Legend: The role of perceived conflict and receptiveness to opposing views in the relationship between zero-sum beliefs and the avoidance of political conversation among U.S. voters (Study 2); $n = 560$.

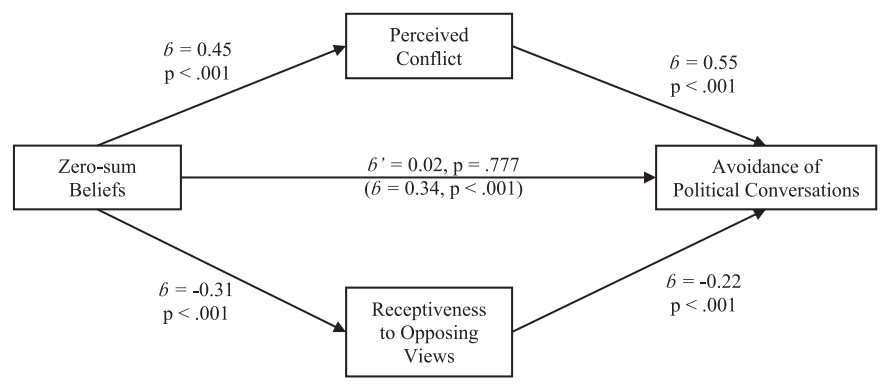

opposing views, which statistically predicted their avoidance of such conversations.

One week later (on the day before the 2022 U.S. midterm elections), 480 participants returned for the second wave of our study (Time 2). Participants indicated their zero-sum beliefs about politics and whether they actively avoided talking about it with ideologically opposed others during the week that passed since their initial participation (i.e., since Time 1).

As before, zero-sum beliefs at Time 2 statistically predicted the avoidance of political conversations ($b = 0.44$, 95% CI[0.25, 0.64], $t(478) = 4.44$, $p < 0.001$), and this was true even when controlling for whether participants identified, as Republicans, Democrats, or Independent ($b = 0.45$, 95% CI[0.25, 0.64], $t(476) = 4.55$, $p < 0.001$; Table 4, Model 1). Zero-sum beliefs also predicted the avoidance of political conversations when controlling for participants' personality traits, self-reported perspective-taking abilities, and a continuous measure of their political ideology (*very liberal* to *very conservative*) ($b = 0.40$, 95% CI[0.20, 0.59], $t(462) = 4.00$, $p < 0.001$; Table 4, Model 2). Finally, a linear regression predicting avoidance of political conversation found no effect of political extremism (operationalized as the absolute distance of participants' liberalism/conservatism from the political ideology scale's midpoint) ($b = 0.05$, 95% CI[−0.13, 0.23], $t(477) = 0.52$, $p = 0.601$; Table 4, Model 3).

As before, speaking to the robustness of these findings, we did not find evidence of moderation of the predictive effect of zero-sum beliefs by political party affiliation (measured as a three-level categorical variable: Republican, Democrat, Independent; Table 5, Model 1) or political

orientation (measured as a continuous, 7-point scale of their liberalism/conservatism; Table 5, Model 2).

We next examined the predictive power of zero-sum beliefs across the two different time points. Two linear regressions examining responses in Time 1 and Time 2 found that the predictive power of zero-sum beliefs at Time 1 on conversational avoidance at Time 2 ($b = 0.45$, 95% CI[0.25, 0.66], $t(478) = 4.30$, $p < 0.001$) was more than 4 times larger than the predictive power of conversational avoidance at Time 1 on zero-sum beliefs at Time 2 ($b = 0.10$, 95% CI[0.06, 0.14], $t(478) = 4.62$, $p < 0.001$). Put differently, participants' baseline zero-sum beliefs at Time 1 better predicted their conversational avoidance at Time 2 than did their avoidance of such conversations at Time 1 predicted their zero-sum beliefs at Time 2. Thus, while the relationship between zero-sum beliefs and the avoidance of political conversations may be bidirectional, these findings suggest that such beliefs have a more pronounced predictive effect on the avoidance political conversations than the reverse.

At the same time, although time-lagged correlations provide a valuable perspective on the potential directionality and temporal sequence of statistical relationships, it is extremely important to approach such findings with caution as other unmeasured factors may influence both zero-sum beliefs and avoidance behavior. As such, our findings underscore the need for a comprehensive understanding of the complex relationship between zero-sum beliefs and conversation avoidance, and emphasize the importance of further research that incorporates experimental and longitudinal designs.

**Table 4 | Study 2, Time 2: Linear regressions predicting avoidance of political conversations from zero-sum beliefs and political party affiliation (three-level categorical variable: Democrat vs. Republican vs. Independent; Model 1), zero-sum beliefs, Big-Five personality traits, self-reported perspective-taking abilities, and political orientation (7-point Likert scale: very liberal—very conservative; Model 2), and zero-sum beliefs and political extremism (a continuous, 4-point scale denoting the absolute distance of their liberalism/conservatism from the midpoint of the political ideology scale; Model 3)**

|  | Predictor | *b* | *p* value | 95% CI | *sr²* | *sr²* 95% CI | Fit |
|---|---|---|---|---|---|---|---|
| Model 1 | Intercept | 2.73 | 3.86e-10 | [1.89, 3.57] | | | |
| | Zero-Sum Beliefs | 0.45 | 6.85e-06 | [0.25, 0.64] | 0.04 | [0.01, 0.07] | |
| | Republican | −0.52 | 0.0009 | [−0.82, −0.21] | 0.02 | [−0.00, 0.05] | |
| | Independent/Other | 0.26 | 0.6513 | [−0.86, 1.38] | 0.00 | [−0.00, 0.00] | |
| | | | | | | | *R²* = 0.063 95% CI[0.02, 0.11] |
| Model 2 | Intercept | 3.16 | 0.0001 | [1.56, 4.76] | | | |
| | Zero-Sum Beliefs | 0.40 | 7.5e-05 | [0.20, 0.59] | 0.03 | [0.00, 0.06] | |
| | Political Orientation | −0.10 | 0.0174 | [−0.17, −0.02] | 0.01 | [−0.01, 0.03] | |
| | Extraversion | −0.21 | 0.0024 | [−0.35, −0.08] | 0.02 | [−0.00, 0.04] | |
| | Conscientiousness | 0.12 | 0.2336 | [−0.08, 0.32] | 0.00 | [−0.01, 0.01] | |
| | Agreeableness | 0.04 | 0.6170 | [−0.13, 0.21] | 0.00 | [−0.00, 0.00] | |
| | Neuroticism | 0.08 | 0.2923 | [−0.07, 0.23] | 0.00 | [−0.01, 0.01] | |
| | Openness | 0.11 | 0.1914 | [−0.05, 0.27] | 0.00 | [−0.01, 0.01] | |
| | Perspective-Taking | −0.22 | 0.0477 | [−0.45, −0.00] | 0.01 | [−0.01, 0.02] | |
| | | | | | | | *R²* = 0.087 95% CI[0.03, 0.12] |
| Model 3 | Intercept | 2.42 | 1.54e-07 | [1.53, 3.31] | | | |
| | Zero-Sum Beliefs | 0.44 | 1.26e-05 | [0.24, 0.63] | 0.04 | [0.01, 0.07] | |
| | Political Extremism | 0.05 | 0.6010 | [−0.13, 0.23] | 0.00 | [−0.00, 0.00] | |
| | | | | | | | *R²* = 0.040 95% CI[0.01, 0.08] |

**Table 5 | Study 2, Time 2: Linear regressions predicting avoidance of political conversations from zero-sum beliefs, political party affiliation, and their interaction (three-level categorical variable: Democrat vs. Republican vs. Independent; Model 1) and zero-sum beliefs, political orientation, and their interaction (7-point Likert scale: very liberal—very conservative; Model 2)**

|  | Predictor | *b* | *p* value | *b* 95% CI | *sr²* | *sr²* 95% CI | Fit |
|---|---|---|---|---|---|---|---|
| Model 1 | Intercept | 2.31 | 0.0005 | [1.01, 3.61] | | | |
| | Zero-Sum Beliefs | 0.55 | 0.0005 | [0.24, 0.85] | 0.02 | [−0.00, 0.05] | |
| | Republican | 0.24 | 0.7827 | [−1.46, 1.93] | 0.00 | [−0.00, 0.00] | |
| | Independent/Other | −1.04 | 0.7794 | [−8.33, 6.25] | 0.00 | [−0.00, 0.00] | |
| | Zero-Sum Beliefs x Republican | −0.18 | 0.3737 | [−0.58, 0.22] | 0.00 | [−0.01, 0.01] | |
| | Zero-Sum Beliefs x Independent/Other | 0.30 | 0.7246 | [−1.39, 2.00] | 0.00 | [−0.00, 0.00] | |
| | | | | | | | *R²* = 0.065 95% CI[0.02, 0.10] |
| Model 2 | Intercept | 2.04 | 0.0368 | [0.13, 3.96] | | | |
| | Zero-Sum Beliefs | 0.65 | 0.0046 | [0.20, 1.10] | 0.02 | [−0.01, 0.04] | |
| | Political Orientation | 0.08 | 0.6878 | [−0.32, 0.48] | 0.00 | [−0.00, 0.00] | |
| | Zero-Sum Beliefs x Political Orientation | −0.04 | 0.3453 | [−0.14, 0.05] | 0.00 | [−0.01, 0.01] | |
| | | | | | | | *R²* = 0.057 95% CI[0.02, 0.10] |

## Discussion

Why do people avoid talking about politics with ideologically opposed others? Two studies conducted on the days and weeks leading up to two highly consequential elections found that both Israeli and American voters tended to avoid political conversations when they saw politics as zero-sum. Such zero-sum beliefs about politics statistically predicted the avoidance of political conversations through two distinct psychological processes: perceived conflict and a lack of receptiveness to opposing views. The more participants saw politics as zero-sum, the more they believed that talking about it creates conflict and the less receptive they were to counter-attitudinal information. As a result, seeing political discourse as an antagonistic battle and being unreceptive to others' views statistically predicted participants' avoidance of political conversations. Thus, in the same way that people avoid negotiations that they see as zero-sum[58], viewing politics as such statistically predicted whether people avoid talking about it with ideologically opposed others.

Our findings are important for understanding people's avoidance of political conversations. By depicting politics as zero-sum[71,72], politicians and political pundits may encourage people to actively avoid opposing views. Similarly, the rise of dominance-prone leaders[73] (who tend to foster zero-sum beliefs among their followers[74]) may cultivate a view of politics as zero-sum. Consequently, such zero-sum beliefs may inhibit political discussions

among ideologically opposed individuals, creating echo chambers and exacerbating existing political divisions[75–83]. Thus, examining how zero-sum beliefs affect the avoidance of political conversations may be critical for understanding political divisions in society.

At the same time, we acknowledge the need for further research into the causal link between zero-sum beliefs and the avoidance of political conversations. Although we provide robust evidence that zero-sum beliefs statistically and temporally predict the avoidance of political conversations (and that this is due to people's fear of conflict and their lack of receptiveness to opposing views), the correlational nature of our studies limits any definitive determination of causality. Thus, we look forward to future research on the topic. By further exploring the underlying psychological mechanisms that lead people to avoid political conversations, future research will not only contribute to our understanding of causality but may also help explore potential interventions to facilitate such conversations.

## Limitations

While we consistently found that zero-sum beliefs were associated with the avoidance of political conversations, we acknowledge the need for further research into the accuracy of such beliefs. Indeed, one promising avenue for future research lies in exploring the distinctions between zero-sum beliefs about political power in general and zero-sum beliefs about specific political legislation. For instance, whereas elections are clearly zero-sum, bipartisan legislation can benefit voters of more than one party and is therefore non-zero-sum[84]. Similarly, while some political actions may be zero-sum (e.g., restrictions that target an opposing party's voters), others may be non-zero-sum (e.g., campaigns that encourage voting regardless of one's politics), thus exposing a potentially fertile topic for future inquiry. Yet, regardless of whether people are right or wrong to view politics as zero-sum, the implications of holding such beliefs may be substantial. Just as viewing status as zero-sum inhibits help-giving and motivates aggression irrespective of its veracity[51,73,85,86], people who view politics as zero-sum may shield themselves from opposing views irrespective of the correctness of their beliefs.

It is similarly critical for future research to examine the potential bi-directionality of the relationship between zero-sum beliefs and the avoidance of political conversations. While zero-sum beliefs may lead people to refrain from engaging in political discourse, it is also possible that persistently abstaining from political conversations increases zero-sum beliefs, fostering a bidirectional relationship between the two constructs[87–89]. If so, future research could examine the potential vicious cycle that emerges, with zero-sum beliefs increasing the avoidance of the opposite side which further strengthens people's zero-sum beliefs and so forth. Moreover, by examining the potential reciprocal nature of these two psychological constructs, future research may further our understanding of prolonged intergroup conflicts and political deadlocks, in which people's perceptions affect who they engage with and, consequently, their subsequent perceptions. Thus, future research could delve into the underlying causal mechanisms at play, warranting a more comprehensive exploration of how these two factors interact and potentially influence each other over time and shedding light on the complex interplay between individual beliefs, communication habits, and the development of political perspectives.

Finally, future research could further explore the distal and proximal causes of zero-sum beliefs about politics which may prove critical for encouraging more political conversations. For instance, since deliberation reduces zero-sum beliefs[31], prompting people to consider how the political arena can benefit large swaths of the electorate may weaken their resistance to interacting with ideologically opposed others. And, since financial vulnerability increases zero-sum beliefs[86,90], relieving financial hardships may similarly reduce the avoidance of such conversations. Thus, future research could motivate political conversations by examining ways to reduce zero-sum beliefs about politics.

The fact that zero-sum beliefs statistically predict the avoidance of political conversations in two distinct political systems is noteworthy. Whereas the political landscape in the U.S. is dominated by an increasingly sectarian two-party system[91], the Israeli multi-party system necessitates coalition-building across parties. Yet, despite these differences (and building on cross-cultural research on the topic[35,92,93]), we found that both Israelis and Americans who see politics as zero-sum avoided talking about it.

It is similarly noteworthy that zero-sum beliefs statistically predicted people's actual avoidance of political conversations rather than their intentions to do so or their beliefs about hypothetical scenarios. Given our interest in real-world conversations, and due to the difficulty of manipulating zero-sum beliefs in ecologically valid manners[31], we focused on measuring (rather than manipulating) participants' zero-sum beliefs about politics. And, while correlational data should always be taken with a grain of salt, the longitudinal findings of Study 2 suggest that zero-sum beliefs better predict the avoidance of political conversations than vice-versa. Future research could test the effect of such beliefs in more controlled settings, examining the causal impact of such beliefs on the avoidance of political conversations.

## Conclusion

The rise of political polarization[94–96] and the ensuing sectarianism and animosity among political partisans[91,97,98] may be associated with the avoidance of meaningful discussions with ideologically opposed others. While research often focuses on changing partisans' perceptions of their political outgroup[8,99–103] and their meta-perceptions of how their group is seen by the outgroup[49,50,104], it has inadvertently overlooked a key aspect of such division: partisans' zero-sum beliefs about the nature of politics. By focusing on how zero-sum beliefs an foster avoidance of political conversations, we hope to make a first step toward encouraging conversations across the political divide. To encourage people to play the game of politics, we may first need to change their beliefs about the nature of the game itself.

## Data availability

The materials and datasets generated for these studies have been deposited in the Open Science Framework (OSF) repository and are accessible here: https://doi.org/10.17605/OSF.IO/FMBER.

## Code availability

The code for the analyses of Studies 1 and 2 is available online through the Open Science Framework: https://doi.org/10.17605/OSF.IO/FMBER.

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

## Author contributions

Shai Davidai designed the materials and conducted the studies. F. Katelynn Boland conducted the data analyses. Both authors contributed equally to writing the manuscript.

## Competing interests

The authors declare no competing interests.
