## [Peer Review File · Communications Psychology]

14th Aug 23

Dear Dr Davidai,

Thank you for your patience during the peer-review process. Your manuscript titled "Zero-Sum Beliefs and the Avoidance of Political Conversations" has now been seen by 3 reviewers, whose comments are appended below. You will see that they find your work of some potential interest. However, they have raised quite substantial concerns that must be addressed. In light of these comments, we cannot accept the manuscript for publication, but would be interested in considering a revised version that fully addresses these serious concerns.

We hope you will find the Reviewers' comments useful as you decide how to proceed. Should additional work allow you to address these criticisms, we would be happy to look at a substantially revised manuscript. If you choose to take up this option, please highlight all changes in the manuscript text file, and provide a detailed point-by-point reply to the reviewers.

In particular, we ask you to focus your revisions on three domains: the treatment of correlational data; suitable control analyses; and conceptual and presentational clarity.

The reviewers highlight that the nature of the study does not allow for a causal interpretation of results. Especially the main effects (of which one is not statistically significant) cannot be interpreted as directional evidence. Any claims of causality or causal language need to be carefully removed from the manuscript. The presentation should make it clear, from the Abstract onward, that the correlational data cannot support causal interpretation; the manuscript should explicitly acknowledge that it remains unclear whether zero-sum beliefs make people refrain from political conversations, whether a habitual lack of political conversations results in more zero-sum beliefs, or whether it could be a bi-directional relationship. Any recommendations for interventions need to be removed from the manuscript. Please also remove the last sentence of the Abstract, which highlights speculation that should be removed from the Discussion. Please include the subheading "Limitations" in your Discussion section to signpost these to the reader.

Second, the reviewers highlight a number of concerns and point out ambiguities regarding the existing analysis; they suggest additional analyses to provide stronger evidence for your key claims, especially those arising from the mediation model. Please address these requests comprehensively. Moreover, as Reviewer #2 highlights, the main effect in the Israeli cohort does not reach conventional levels of statistical significance (although the analysis that controls for political leaning does). Please note that according to journal guidelines, marginally statistically significant effects may (and should) be reported but must not be interpreted. As such, the main effect in the first study cannot be described as confirming the hypothesis (you hypothesized an association, but the evidence is ambiguous).

Third, all reviewers, and especially Reviewer #1, point to a lack of conceptual clarity and definitions. This criticism should be addressed through comprehensive textual revisions. In the same vein, although your manuscript is considered for inclusion in the "Polarization" collection, it does not

measure polarization; the links to polarization should therefore be toned down (see comments above regarding the Abstract and the discussion of interventions). These textual changes will not stand in the way of having your paper included in the Collection

If the revision process takes significantly longer than 12 weeks, we will be happy to reconsider your paper at a later date, provided it still presents a significant contribution to the literature at that stage.

Please use the following link to submit your revised manuscript, point-by-point response to the Reviewers' comments with a list of your changes to the manuscript text (which should be in a separate document to any cover letter) and any completed checklist:

[link redacted]

Please do not hesitate to contact me if you have any questions or would like to discuss the required revisions further. Thank you for the opportunity to review your work.

Best regards,

Antonia Eisenkoeck

Antonia Eisenkoeck
Senior Editor
Communications Psychology

EDITORIAL POLICIES AND FORMATTING

Editorial Policy: Policy requirements (Download the link to your computer as a PDF.)

Furthermore, please align your manuscript with our format requirements, which are summarized on the following checklist:

Communications Psychology formatting checklist

and also in our style and formatting guide Communications Psychology formatting guide .

* **CODE AVAILABILITY:** All Communications Psychology manuscripts must include a section titled "Code Availability" at the end of the methods section. In the event of publication, we require that the custom analysis code supporting your conclusions is made available in a publicly accessible repository; please choose a repository that provides a DOI for the code; the link to the repository and the DOI must be included in the Code Availability statement. Publication as Supplementary Information will not suffice. We ask you to prepare and upload code at this stage, to avoid delays later on in the process.

* **DATA AVAILABILITY:**

All Communications Psychology research manuscripts must include a section titled "Data Availability" at the end of the Methods section or main text (if no Methods). More information on this policy, is available at <http://www.nature.com/authors/policies/data/data-availability-statements-data-citations.pdf>.

At a minimum the Data availability statement must explain how the data can be obtained and whether there are any restrictions on data sharing. Communications Psychology strongly endorses open sharing of data. If you do make your data openly available, please include in the statement:

We recommend submitting the data to discipline-specific, community-recognized repositories, where possible and a list of recommended repositories is provided at <http://www.nature.com/sdata/policies/repositories>.

If a community resource is unavailable, data can be submitted to generalist repositories such as figshare or Dryad Digital Repository. Please provide a unique identifier for the data (for example a DOI or a permanent URL) in the data availability statement, if possible. If the repository does not provide identifiers, we encourage authors to supply the search terms that will return the data. For data that have been obtained from publicly available sources, please provide a URL and the specific data product name in the data availability statement. Data with a DOI should be further cited in the methods reference section.

REVIEWER EXPERTISE:

Reviewer #1: political/economic beliefs, statistics

Reviewer #2: conflict avoidance

Reviewer #3: political/economic beliefs

Reviewer #1 (Remarks to the Author):

Review of COMMSPSYCHOL-23-0189-T

“Zero-Sum Beliefs and the Avoidance of Political Conversations”

- Unless I missed it, “zero-sum” is never defined (I understand what the authors mean, but the definition can help expand readership and facilitate a clear understanding of the paper)
- Do the authors expect any ideological asymmetries in the relationships they hypothesize? It seems from past work that there may be reason to (<https://www.science.org/doi/10.1126/sciadv.aay3761>)
- Relatedly, the paper seems to conflate partisanship and ideology. Ideology is not asked about in either sample, only partisanship.
- “Thus, by investigating why people avoid political conversations, we hope to advance the literature on polarization and offer a critical step toward interventions that reduce such avoidance.” This sentence gets to a major theoretical issue: isn’t zero-sum thinking, itself, a product of polarization, rather than a cause? Do the authors have access to feeling thermometer ratings of the parties/candidates or attitudes about political issues? If so, it would be nice to see whether zero-sum thinking is related to affective/ideological polarization (and, even better, the causal ordering). Without this, the paper feels fairly circular: people who are prone to disliking and avoiding the out-party see politics as a zero-sum fight...or maybe it’s the other way around? Or maybe the relationships are reciprocal (as shown in the paper)? Or maybe we simply cannot disentangle any causal ordering at this highly polarized time in 2023?
- Ultimately, the paper makes only a modest contribution to our understanding of political behavior. The conclusion is akin to something like: “People who perceive political conflict are most likely to avoid political conflict.” This just makes sense—one can’t avoid conflict unless they believe it to be lurking around the corner. The modest contribution is especially so given the reciprocal effects of zero-sum beliefs on conversational avoidance (and vice versa) across time.

Reviewer #2 (Remarks to the Author):

In the current manuscript, entitled “Zero-Sum Beliefs and the Avoidance of Political Conversations”, the authors examined the hypothesized role of zero-sum beliefs as a predictor of avoidance of politically charged conversations. The authors found in two studies conducted in Israel and the U.S. that zero-sum beliefs were indeed negatively associated with engaging in politically-charged conversations and that this was mediated by perceived conflict and receptiveness to opposing views. I enjoyed reading the manuscript as it was clearly written, and deals with what I think is an important topic, namely, political polarization. Furthermore, I liked the focus on real-world relevance and the fact that the surveys were conducted in close proximity to elections in both contexts. However, I believe that there are some major issues that should be addressed to make it a substantial contribution to the literature. I list these issues below, and I hope that the authors would

find them useful.

1. I think that the authors should refrain from conflating political discussions with politically opposed others and political discussions on politically charged topics. I think that these two, albeit similar, are different things, that can be predicted by different variables (e.g., argumentativeness, willingness to self-censor) and can have different outcomes.

2. I think that it's important to note several potential predictors for why people avoid political conversations before going into zero-sum perceptions. I mentioned above a couple of personality characteristics, similar to zero-sum perceptions, but I think that it's important to note that it could also be contextual. One of the argument of intergroup conflict researchers is that zero-sum perceptions are one of the main characteristics of intergroup conflicts (especially prolonged and violent ones; e.g., the work of Kriesberg and Bar-Tal). Thus, an argument could be made that the polarization processes that are taking place in many places around the world, and in particular in the U.S. and Israel, essentially shifted how people perceive the inter-partisan relations in their country to that akin to intergroup conflict, which leads to this zero-sum perception of the relations. In other words, what I am trying to ask here is whether avoiding political conversations is the result of zero-sum beliefs, or the fact that the inter-partisan relations are perceived as an intergroup conflict, with zero-sum beliefs as just one manifestation of that.

3. I am not sure I managed to follow the argument about the first potential mediator, i.e., that seeing politics as zero-sum leads to worrying that talking about it creates conflict and hostility. Zero-sum beliefs make animosity seem desirable, so people avoid it? Following the argument made on p. 4, lines 88-91, I would hypothesize that zero-sum beliefs would actually increase the willingness to engage in conflictual discussions. I would understand the argument more if zero-sum beliefs would only be associated with heightened perceptions of conflict. But, if it also makes conflict more normative and desirable, then the argument makes less sense to me. I would argue that avoiding negotiations is somewhat different, as zero-sum beliefs would also be associated with the perception that engaging in negotiation is a futile endeavor (which is similar to the second mediator, to some extent). The second mediator about receptiveness of counter-attitudinal information makes more sense to me.

4. At the end of the introduction, the authors write: "In sum, we predict that zero-sum beliefs about politics can help explain why many people avoid political conversations with ideologically opposed others. We argue that people who view politics as zero-sum are less receptive to opposing views and expect political conversations to be conflict-prone which, as a result, explains why they avoid talking about it." After reading the introduction, I am still not completely convinced by the argument that zero-sum beliefs *lead* to perceived conflict, and it's not the other way around (or at least a manifestation of conflict, and not something novel), as intergroup conflict scholars would argue. Also, if the main predictor is zero-sum beliefs, then I think it'll be good to elaborate a bit more on what is it – is it a personality trait? Mindset? As well as what predicts zero-sum beliefs.

5. I think that it's not ideal at all that the main result of the paper is marginally significant in the Israeli sample and should be discussed (especially if it was not preregistered, which is the case, as far as I can understand). It doesn't seem like a statistical power issue.

6. I am not sure how political identification was measured in Study 1. Was it with a 1 = extreme right to 7 = extreme left measure? Or was it measured based on voting to political parties? It's not clear. If

the latter, since it is a categorical measure, how was the analysis conducted? This is particularly important given the marginal result that becomes significant after controlling for political affiliation and the fact that this analysis was not preregistered, as far as I can tell. Furthermore, it'll be interesting to examine whether political affiliation moderates the effects.

7. It'll be good to elaborate more on the measures (e.g., personality traits, political affiliation) in Study 2. It's not clear what those were and the rationale for including them. Is there any research that suggests that perspective taking is associated with avoidance from politically charged conversations? Furthermore, I am curious here as well whether political affiliation moderated the results.

8. Although the authors address the fact that the studies are correlational, I think that given that the main argument of the research is the **causal** effect of zero-sum beliefs, and given what I wrote above about the research on intergroup conflict and how zero-sum perception is a manifestation of that, this warrants more discussion. Longitudinal assessments may alleviate some concerns about directionality, but they do not establish it (relevant references include: Maxwell & Cole, 2007; Rohrer, Hünermund, Arslan, & Elson, 2022; Selig & Little, 2012). It goes without saying that an experiment is of course the ideal way to establish causality.

Minor issues:

9. On p. 13, line 285, what type of effect is examined in the power analysis? Correlation? Regression coefficient? (Similar question is relevant to Study 2)

10. What was the attention check used in Study 2.

Reviewer #3 (Remarks to the Author):

This paper looks at zero-sum thinking related to politics, arguing that those higher in this trait are more prone to avoiding political conversations. The authors present two survey studies – one longitudinal – demonstrating a correlational relationship between these variables.

I think this paper makes a solid contribution to the literature on zero-sum thinking and political psychology. I have only a few suggestions for improving the paper further.

1. The only substantial issue I see is that zero-sum thinking may be confounded with the **extremity** (rather than liberal/conservative orientation) of political ideology. If more extreme partisans (of either type) are more prone to zero-sum thinking and are also more prone to avoid political conversations, this could undermine the authors' interpretations of the results. Thus, I strongly recommend that the authors include models that adjust not only for the valence of political beliefs but for their extremity (e.g., absolute value from the scale midpoint). I would like to see this done both for the simple direct effects models as well as for the mediation models (i.e., including political beliefs as a covariate).

2. As a more minor issue, I wasn't entirely clear on why it is important that the studies be carried out in the days leading up to an election. Is the argument just that this is a particularly consequential

time for political discussions, or is there some reason why the effect would be moderated by proximity to an election?

3. The authors seem to view the time-lagged correlation evidence from Study 2 as more indicative of causal directionality compared to the other evidence reported in the paper, but wisely recommend caution in interpreting the results. Could the authors please elaborate more on their favored interpretation of this evidence and more thoroughly discuss the limitations of this approach?

4. As a discussion point, I wonder what the authors think about a different kind of zero-sum belief about politics. In their actual questions, the scale measures the zero-sum nature of political power and the success of different political parties (e.g., the idea that if the Republicans are in charge, Democrats will wield proportionally less influence, etc.). A different kind of zero-sum thinking about politics could be about the zero-sum nature of government policy (e.g., taxation, redistribution, regulation, etc.). Do the authors think that zero-sum beliefs about political power and zero-sum beliefs about policy would have similar predictive effects on political behavior?

Related minor point – in their paragraph about the accuracy of zero-sum political views in the Discussion, the authors make it sound like their scale is measuring both of these kinds of political beliefs (both about political power and legislation), but I don't think the scale is really tapping into the latter. I would suggest rephrasing this paragraph to avoid this implication (perhaps framing it around a future direction would be a more accurate way to make the same point).

Response to Reviews

We are very thankful for the supportive and insightful reviews by the three anonymous reviewers who highlighted many positive aspects of the manuscript and thus strengthened our belief in the novelty and importance of our work. We are especially grateful for the reviewers' comments which emphasized the paper's "*real-world relevance*," its examination of "*an important topic*," and their belief that the paper "*makes a solid contribution to the literature on zero-sum thinking and political psychology*."

Of course, the review team also raised some very helpful and constructive comments that helped us strengthen the manuscript. We wholeheartedly agree with these comments and view the reviewers' attention to detail as an indication that this is a worthwhile and important research project. We therefore devoted a considerable amount of time to the reviews and ran additional analyses, edited the manuscript, and clarified our work's contribution to the literature, all of which have helped us to significantly enhance the paper's theoretical and empirical foundations.

Our objective is to make our paper as theoretically and empirically robust as possible and we thank the Editor and reviewers for urging us to revise the manuscript and for giving us the opportunity to do so. Below, we outline each of the questions and comments that were brought up during the review process and describe in detail how we addressed them. As a result, we believe that the manuscript is substantially stronger and more impactful.

Reviewer 1

“Unless I missed it, “zero-sum” is never defined (I understand what the authors mean, but the definition can help expand readership and facilitate a clear understanding of the paper)”

We completely agree that adding a clear definition of zero-sum beliefs is important and can help facilitate a greater understanding of the paper. Therefore, in revising the manuscript, we have made sure to do so. Specifically, building upon a recent review of the topic, we now include the following definition of zero-sum beliefs in the revised manuscript: “the implicit or explicit belief that any gain made by one party must result in an equivalent loss for another party”. This definition can be found on page 5 of the revised Introduction.

“Do the authors expect any ideological asymmetries in the relationships they hypothesize? It seems from past work that there may be reason to”

While previous work (including our own research) has found ideological asymmetries in whether and how people exhibit zero-sum beliefs, it is important to note that the current studies examined the potential impact of such beliefs rather than people’s tendency to exhibit them in the first place. Thus, while there are ideological differences in what people see as zero-sum, we did not expect any political or ideological differences in the effect of such beliefs on people’s thoughts and behaviors. Indeed, while the lack of significance should always be taken with a grain of salt, it is important to note that neither political ideology nor political affiliation moderated the predictive power of zero-sum beliefs in the reported studies. Thus, while ideological

asymmetries may indeed exist in people's tendency to exhibit zero-sum beliefs, ideology does not seem to moderate the effect of such beliefs on people's avoidance of political conversation.

“Relatedly, the paper seems to conflate partisanship and ideology. Ideology is not asked about in either sample, only partisanship.”

We completely agree with this comment and have made sure to correct this issue in the revised manuscript. Specifically, as stated by Reviewer 1, the first study in the manuscripts examined political affiliation (i.e., which political party they support) which does not necessarily reflect participants' political ideology. In contrast, Study 2 measured both political affiliation (i.e., whether participants identify as Democrat, Republican, or Independent) as well as ideology (on a conservative-liberal scale). In revising the manuscript, we made sure to explicitly state that we only measure political affiliation in Study 1 and both aspects in Study 2. We thank Reviewer 1 for this feedback, as it allowed us to refine the way we describe the studies in the paper.

“Thus, by investigating why people avoid political conversations, we hope to advance the literature on polarization and offer a critical step toward interventions that reduce such avoidance.’ This sentence gets to a major theoretical issue: isn't zero-sum thinking, itself, a product of polarization, rather than a cause? Do the authors have access to feeling thermometer ratings of the parties/candidates or attitudes about political issues? If so, it would be nice to see whether zero-sum thinking is related to affective/ideological polarization (and, even better, the causal ordering). Without this, the paper feels fairly circular: people who are prone to disliking and avoiding the out-party see politics as a zero-

sum fight...or maybe it's the other way around? Or maybe the relationships are reciprocal (as shown in the paper)? Or maybe we simply cannot disentangle any causal ordering at this highly polarized time in 2023?"

Reviewer 1 correctly noted that the paper does not establish a causal relationship between zero-sum beliefs and the avoidance of political conversations. And, while Study 2 finds robust evidence that zero-sum beliefs are more strongly predictive of such avoidance one week later than vice-versa, such longitudinal data suggests (but does not establish) the potential causal relationship between the two constructs. Indeed, as we note in the revised manuscript, we completely agree that the relationship between these two constructs may be bi-directional, creating a vicious cycle between zero-sum beliefs and the avoidance of political conversations. Yet, while we concede that future research can benefit from pinpointing the causal link between these two constructs, we are confident that the current findings offer an important contribution to the literature. Specifically, highlighting the relationship between zero-sum beliefs (i.e., people's view of *the nature* of the political realm) and the avoidance of political conversation is important for advancing our understanding of why people fail to engage with ideologically opposed others. Nevertheless, Reviewer 1's comment is well warranted, and we have made sure to explicitly note in the revised manuscript that the relationship between these two constructs may be reciprocal and might lead to a vicious cycle where zero-sum beliefs increase avoidance of the opposite side which then leads to more zero-sum beliefs and so forth (see page 15). In addition, in revising the manuscript we made sure to remove any inadvertent causal language from the paper. We thank Reviewer 1 for urging us to include this discussion, which we believe paints a more complete picture of the findings.

“Ultimately, the paper makes only a modest contribution to our understanding of political behavior. The conclusion is akin to something like: ‘People who perceive political conflict are most likely to avoid political conflict.’ This just makes sense—one can’t avoid conflict unless they believe it to be lurking around the corner. The modest contribution is especially so given the reciprocal effects of zero-sum beliefs on conversational avoidance (and vice versa) across time.”

As noted above, we respectfully disagree with Reviewer 1’s views of the manuscript’s potential contribution. To be clear, viewing politics as zero-sum is *inherently different* from merely seeing it as a two-sided conflict. Whereas many two-sided conflicts can be potentially resolved through ‘win-win’ solutions that benefit all involved parties, zero-sum conflicts are unique in the sense that they cannot be solved in such a manner. Rather, zero-sum conflicts are ones in which each party’s gains are inevitably accrued at the expense of another party’s losses. Thus, viewing politics as zero-sum specifically relates to the belief that the political sphere is a *unique* kind of conflict *in which people can only gain at others expense*. While conflicts in general can involve both types of situations—zero-sum and non-zero-sum—viewing politics as zero-sum is a specific kind of conflict.

In revising the manuscript, we have made sure to clarify this issue and explicitly note that zero-sum beliefs about politics are *not* synonymous with merely viewing politics as a conflict. In addition, we have made sure to include additional citations to the manuscript that specifically relate to the ‘conflict aversion’ literature, which show that people vary in their aversion to

conflict. The discussion about the specific nature of *zero-sum* conflicts can be found on page 6 of the revised manuscript.

Reviewer 2

“I enjoyed reading the manuscript as it was clearly written, and deals with what I think is an important topic, namely, political polarization. Furthermore, I liked the focus on real-world relevance and the fact that the surveys were conducted in close proximity to elections in both contexts. However, I believe that there are some major issues that should be addressed to make it a substantial contribution to the literature. I list these issues below, and I hope that the authors would find them useful.”

We wholeheartedly thank Review 2 for this feedback! We are extremely happy to hear that the reviewer enjoyed the paper and found it important for real-world issues.

“I think that the authors should refrain from conflating political discussions with politically opposed others and political discussions on politically charged topics. I think that these two, albeit similar, are different things, that can be predicted by different variables (e.g., argumentativeness, willingness to self-censor) and can have different outcomes.”

We thank Reviewer 2 for this comment, which we admit has not previously occurred to us. We completely agree that the paper ought not conflate *politically opposed others* with *politically*

charged topics and, in revising the manuscript, have made sure to avoid doing so. Specifically, while we believe that studying politically charged topics is both interesting and important in and of itself, we make clear in the revised manuscript that we particularly focus on avoiding political discussions with politically opposed others.

“I think that it’s important to note several potential predictors for why people avoid political conversations before going into zero-sum perceptions. I mentioned above a couple of personality characteristics, similar to zero-sum perceptions, but I think that it’s important to note that it could also be contextual. One of the arguments of intergroup conflict researchers is that zero-sum perceptions are one of the main characteristics of intergroup conflicts (especially prolonged and violent ones; e.g., the work of Kriesberg and Bar-Tal). Thus, an argument could be made that the polarization processes that are taking place in many places around the world, and in particular in the U.S. and Israel, essentially shifted how people perceive the inter-partisan relations in their country to that akin to intergroup conflict, which leads to this zero-sum perception of the relations. In other words, what I am trying to ask here is whether avoiding political conversations is the result of zero-sum beliefs, or the fact that the inter-partisan relations are perceived as an intergroup conflict, with zero-sum beliefs as just one manifestation of that.”

Reviewer 2 is absolutely correct that there are several potential predictors for why people avoid political conversations. Indeed, we completely agree that discussing these predictors can greatly benefit the paper and, given that there are many factors that contribute to people avoiding political conversations, it is crucial to consider the broader context in which this phenomenon

occurs. Thus, in revising the manuscript, we made sure to add a paragraph to the Introduction in which we discuss various personality characteristics, cognitive process, and contextual and social-cultural factors that may contribute to people's avoidance of political conversations. We believe that adding this discussion helps better situate our research within the larger literature and theoretical framework and thank Reviewer 2 for making this suggestion and bringing this to our attention.

“I am not sure I managed to follow the argument about the first potential mediator, i.e., that seeing politics as zero-sum leads to worrying that talking about it creates conflict and hostility. Zero-sum beliefs make animosity seem desirable, so people avoid it? Following the argument made on p. 4, lines 88-91, I would hypothesize that zero-sum beliefs would actually increase the willingness to engage in conflictual discussions. I would understand the argument more if zero-sum beliefs would only be associated with heightened perceptions of conflict. But, if it also makes conflict more normative and desirable, then the argument makes less sense to me. I would argue that avoiding negotiations is somewhat different, as zero-sum beliefs would also be associated with the perception that engaging in negotiation is a futile endeavor (which is similar to the second mediator, to some extent). The second mediator about receptiveness of counter-attitudinal information makes more sense to me.”

In retrospect, we agree that the previous way in which we described the examined psychological mechanism may have not been sufficiently clear and we thank Reviewer 2 for giving us the opportunity to clarify this issue. Specially, we hypothesize in the paper that zero-sum beliefs

about politics lead people to see conflict as an inevitable outcome of talking about it and, consequently, avoid such discussions. Importantly, we do not mean to imply that zero-sum beliefs make conflict seem more *desirable*. Rather, we argue that zero-sum beliefs make conflict seem like an extremely prevalent aspect of political conversations and thus very difficult to avoid. Building upon previous research that has shown how zero-sum beliefs foster a view of negotiations as hostile and conflict-prone, we argue that zero-sum beliefs about politics make people view conversations about it as inevitably riddled with conflict and animosity and, as a result, people try to avoid them. Thus, in revising the manuscript, we have made sure to clarify this point (see page 6 in the revised introduction). Our goal, of course, is to make sure that the paper is as clear as possible and we would thus appreciate further feedback on how to further clarify this point.

“At the end of the introduction, the authors write: “In sum, we predict that zero-sum beliefs about politics can help explain why many people avoid political conversations with ideologically opposed others. We argue that people who view politics as zero-sum are less receptive to opposing views and expect political conversations to be conflict-prone which, as a result, explains why they avoid talking about it.” After reading the introduction, I am still not completely convinced by the argument that zero-sum beliefs *lead* to perceived conflict, and it’s not the other way around (or at least a manifestation of conflict, and not something novel), as intergroup conflict scholars would argue. Also, if the main predictor is zero-sum beliefs, then I think it’ll be good to elaborate a bit more on what is it – is it a personality trait? Mindset? As well as what predicts zero-sum beliefs.”

Reviewer 2 brings up two extremely important points, both of which we have made sure to address in revising the manuscript. First, as noted above, we realize that the previous version of the manuscript may have inadvertently misrepresented our findings and, in revising the paper, have therefore made sure to remove any references to causality. Indeed, as noted by Reviewer 2 (and as we now explicitly discuss in the Discussion) the relationship between these two constructs may be bi-directional and may thus create a vicious cycle between zero-sum beliefs and the avoidance of political conversations. As such, our paper contributes to the literature by highlighting an important and overlooked relationship between zero-sum beliefs, perceived conflict, receptiveness to opposing views, and the avoidance of political conversation. In doing so, we hope that our findings will open up many important avenues for future research.

Second, as noted by Reviewer 2, it's important to clearly conceptualize and operationalize zero-sum beliefs. Therefore, in revising the manuscript we have done exactly that. Specifically, as noted above, we made sure to include in the revised manuscript a clear and explicit definition of zero-sum beliefs. In addition, in revising the manuscript we made sure to clearly state that we conceptualize zero-sum beliefs about politics as a malleable and context dependent mindset that may be affected by the environment in which people find themselves. Indeed, we see the fact that zero-sum beliefs can be affected by people's environment as an indication that their effect is not inevitable but rather something that can be examined and studied within the context of political polarization. The definition of zero-sum beliefs and clarification of the concept can be found on page 5 of the Introduction of the revised manuscript.

“I think that it’s not ideal at all that the main result of the paper is marginally significant in the Israeli sample and should be discussed (especially if it was not preregistered, which is the case, as far as I can understand). It doesn’t seem like a statistical power issue.”

Although the predictive effect of zero-sum beliefs in Study 1 on the avoidance of political conversation is significant when controlling for participants’ political affiliation (an obviously relevant construct), Reviewer 2 is correct that the zero-order correlation between the two constructs was only marginally significant in this study ($p = .054$). In fact, the fact that this relationship was only marginally significant in Study 1 was part of our motivation to conduct Study 2, in which we found a highly significant zero-order correlation between zero-sum beliefs and the avoidance of political conversation (as well as a significant predictive effect of zero-sum beliefs when controlling for political affiliation). As such, we can only hypothesize on why the relationship was only marginally significant in Study 1. For instance, it is possible that because the political system in Israel at the time of the study consisted of 13 different political parties, the effect of zero-sum beliefs (which assumes two inherently conflicting sides) may have been less pronounced. That being said, we completely agree with the reviewer’s concern regarding inadvertently overstating the results and, in revising the manuscript, made sure to avoid doing so.

“I am not sure how political identification was measured in Study 1. Was it with a 1 = extreme right to 7 = extreme left measure? Or was it measured based on voting to political parties? It’s not clear. If the latter, since it is a categorical measure, how was the analysis conducted? This is particularly important given the marginal result that becomes significant after controlling for political affiliation and the fact that this analysis was not

preregistered, as far as I can tell. Furthermore, it'll be interesting to examine whether political affiliation moderates the effects.”

Reviewer 2 raises a great point in their comment, and we thank them for giving us the opportunity to clarify this issue. To be clear, in Study 1 we measured political identification as participants' affiliation with one of 13 political parties in the Israeli election. In Study 2, we measured political identification in two different ways: both as political affiliation (Republican, Democrat, or Independent) and as political ideology (a continuous measure between extreme conservatism and extreme liberalism). To clarify this issue, we made sure to add to the Methods section of the revised manuscript a discussion of these measures. In addition, as suggested by Reviewer 2, we have also made sure to conduct additional analyses that substitute the political affiliation measure in Study 2 with the measure of political ideology. We hope that these edits and additions will help the readers better interpret the results reported in the manuscript.

“It'll be good to elaborate more on the measures (e.g., personality traits, political affiliation) in Study 2. It's not clear what those were and the rationale for including them. Is there any research that suggests that perspective taking is associated with avoidance from politically charged conversations? Furthermore, I am curious here as well whether political affiliation moderated the results.”

We completely agree that further elaborating on the measures used in Studies 1 and 2 can help the readers better interpret the reported results . To be clear, we included in Study 2 measures of

perspective taking and basic personality traits because these have been shown in the past to be strongly associated with people's receptiveness to opposing views – a point we now explicitly clarify in the revised manuscript (see page 18). Since these constructs may affect people's avoidance of political conversations, we made sure to include them in our studies. Of course, it is important to note that the relationship between zero-sum beliefs and the avoidance of political conversations in Study 2 remained significant regardless of whether we controlled for these constructs, thus revealing the robustness of the reported results. Finally, following Reviewer 2's advice, we conducted additional analyses to examine whether the reported results are moderated by participants' political affiliation. As we now report in the Supplementary Materials, these additional analyses did not find any evidence of moderation by political affiliation, suggesting that zero-sum beliefs predict the avoidance of political conversations irrespective of a person's political affiliation.

“Although the authors address the fact that the studies are correlational, I think that given that the main argument of the research is the *causal* effect of zero-sum beliefs, and given what I wrote above about the research on intergroup conflict and how zero-sum perception is a manifestation of that, this warrants more discussion. Longitudinal assessments may alleviate some concerns about directionality, but they do not establish it (relevant references include: Maxwell & Cole, 2007; Rohrer, Hünermund, Arslan, & Elson, 2022; Selig & Little, 2012). It goes without saying that an experiment is of course the ideal way to establish causality.”

As noted above, we completely agree with this comment and, in revising the manuscript, have made sure to remove any references to causal language. In addition, while the longitudinal findings in Study 2 provide robust evidence that zero-sum beliefs at Time 1 much better predict the avoidance of political conversations at Time 2 (one week later) than vice-versa, we have made sure to clarify in the manuscript the limitations of such longitudinal designs. Thank you.

“On p. 13, line 285, what type of effect is examined in the power analysis? Correlation? Regression coefficient? (Similar question is relevant to Study 2)”

Thank you for giving us the opportunity to make this clearer. The power analyses that are reported throughout the paper are post-hoc sensitivity power analyses, examining that smallest detectable effect sizes with 80% power given the achieved sample sizes. In revising the manuscript, we have made sure to clarify this.

“What was the attention check used in Study 2?”

Participants in Study 2 completed an open-ended attention check that asks them to write the number of letters that appear in the word “computer”. We made sure to exclude any responses that differed from ‘8’ or ‘eight.’ To clarify this, we now make sure to note this attention check in the revised manuscript.

Reviewer 3

“I think this paper makes a solid contribution to the literature on zero-sum thinking and political psychology. I have only a few suggestions for improving the paper further.”

Thank you so much for your suggestions, which we have taken as an opportunity to improve the paper. We were extremely thankful to learn that you view this paper as making a solid contribution to the literature and we greatly appreciate this feedback.

“The only substantial issue I see is that zero-sum thinking may be confounded with the *extremity* (rather than liberal/conservative orientation) of political ideology. If more extreme partisans (of either type) are more prone to zero-sum thinking and are also more prone to avoid political conversations, this could undermine the authors’ interpretations of the results. Thus, I strongly recommend that the authors include models that adjust not only for the valence of political beliefs but for their extremity (e.g., absolute value from the scale midpoint). I would like to see this done both for the simple direct effects models as well as for the mediation models (i.e., including political beliefs as a covariate).”

We are extremely thankful for this suggestion, which has not previously occurred to us. To address this issue, we made sure to conduct additional analyses that particularly look at the effect of political *extremity* on the avoidance of political conversations. Unfortunately, since we only measured political ideology in Study 2 (but not in Study 1, in which we measured political affiliation), we were only able to conduct this analysis for this study. Specifically, this analysis found that the extremity of political ideology (operationalized as the absolute distance of participants’ ideology from the scale’s midpoint), did not statistically predict the avoidance of

political conversation ($p = .36$). In contrast, the predictive effect of zero-sum beliefs on the avoidance of political conversation remained highly significant ($p = .00016$) even when controlling for participants' political extremity. Moreover, as an indication of the robustness of these results, we found a similar pattern both when analyzing the first wave of data collection (Time 1) as well as the second wave of data collection (Time 2) of Study 2.

“As a more minor issue, I wasn’t entirely clear on why it is important that the studies be carried out in the days leading up to an election. Is the argument just that this is a particularly consequential time for political discussions, or is there some reason why the effect would be moderated by proximity to an election?”

Great question! To be clear, we decided to run Studies 1 and 2 in the days leading up to the U.S. and Israeli elections out of a belief that political conversations (and political issues in general) are likely to be particularly salient during such times. In fact, such discussions are likely to be especially consequential in the days and weeks leading up to an election, when people are likely to be most engaged with the topic. Thus, in revising the manuscript, we have made sure to clarify this point (page 4).

“The authors seem to view the time-lagged correlation evidence from Study 2 as more indicative of causal directionality compared to the other evidence reported in the paper, but wisely recommend caution in interpreting the results. Could the authors please elaborate more on their favored interpretation of this evidence and more thoroughly discuss the limitations of this approach?”

As noted above, we did not mean to inadvertently represent the results of this longitudinal design as a clear-cut indication of causality but rather as an important sign of the potential causal link that may exist between zero-sum beliefs and the avoidance of political conversation. To be clear, we view this design as providing a valuable (albeit limited) perspective on the potential causal relationship between the two constructs. Specifically, the fact that the relationship between zero-sum beliefs at Time 1 and the avoidance of conversations one week later (at Time 2) is substantially larger than the opposite relationship is, in our opinion, very telling. Nevertheless, to avoid any misrepresentation of our findings, we made sure to edit the manuscript to clarify the limitations of such longitudinal designs (see page 15 of the revised manuscript).

“As a discussion point, I wonder what the authors think about a different kind of zero-sum belief about politics. In their actual questions, the scale measures the zero-sum nature of political power and the success of different political parties (e.g., the idea that if the Republicans are in charge, Democrats will wield proportionally less influence, etc.). A different kind of zero-sum thinking about politics could be about the zero-sum nature of government policy (e.g., taxation, redistribution, regulation, etc.). Do the authors think that zero-sum beliefs about political power and zero-sum beliefs about policy would have similar predictive effects on political behavior?”

Thank you for posing this question, which we found very interesting. While our studies have focused specifically on zero-sum beliefs about political *power*, we agree that examining the role of zero-sum beliefs about political *policies* may be similarly fruitful. Importantly, as we note in

the Discussion section of the revised manuscript (page 13-14), we view the distinction between the two types of zero-sum beliefs as especially important. Specifically, we note how people may have different zero-sum beliefs about different policies, some of which may be indeed zero-sum whereas others may not be zero-sum and may actually benefit the electorate of multiple parties. Thus, while the scope of the current paper required us to focus on only one type of zero-sum beliefs, we view the distinction between the two types of zero-sum beliefs about politics as extremely interesting and one that should definitely be explored in future research.

“Related minor point – in their paragraph about the accuracy of zero-sum political views in the Discussion, the authors make it sound like their scale is measuring both of these kinds of political beliefs (both about political power and legislation), but I don’t think the scale is really tapping into the latter. I would suggest rephrasing this paragraph to avoid this implication (perhaps framing it around a future direction would be a more accurate way to make the same point).”

As suggested, in revising the manuscript we made sure to rephrase this paragraph. We agree that we would like to avoid this type of implication and thank you for pointing this out.

**

Once again, we greatly appreciate the valuable feedback provided by the editor and reviewers and, in revising the manuscript, have made sure to address all comments and suggestions. We believe that these revisions enhance the clarity and robustness of our paper and look forward to further discussion and feedback as we continue to advance our understanding of political behavior in a polarized world.

10th Jan 24

Dear Dr Davidai,

Thank you for your patience during the peer-review process. Your manuscript titled "Zero-Sum Beliefs and the Avoidance of Political Conversations" has now been seen by 2 reviewers, and I include their comments at the end of this message. Whereas one of the reviewers is satisfied with your revisions, the other reviewer raises some outstanding concerns, which we hope you will be able to address through revisions before we make a final decision.

We therefore invite you to revise and resubmit your manuscript, along with a point-by-point response to the reviewers. Please highlight all changes in the manuscript text file.

Specifically, we ask you to address Reviewer #2's methodological concern about the regression analysis, as well as confirmation of some confidence intervals.

Please note that editorially, we discourage claims of causality. However, you may explain in the limitations section that while you would argue that the relationship is likely directional, you acknowledge that the type of data and analysis available limit your ability to test for a causal relationship.

To enable us to make a final decision on your manuscript as efficiently as possible, please also ensure your manuscript complies with our editorial and policy guidelines. I have attached a checklist, which we ask you to use for your revisions.

Please use the following link to submit your revised manuscript, point-by-point response to the referees' comments (which should be in a separate document to any cover letter) and the completed checklist:
[link redacted]

Please do not hesitate to contact me if you have any questions or would like to discuss these

revisions further. We look forward to seeing the revised manuscript and thank you for the opportunity to review your work.

Best regards,

Antonia Eisenkoeck

Antonia Eisenkoeck
Senior Editor
Communications Psychology

* TRANSPARENT PEER REVIEW: Communications Psychology uses a transparent peer review system. This means that we publish the editorial decision letters including Reviewers' comments to the authors and the author rebuttal letters online as a supplementary peer review file. However, on author request, confidential information and data can be removed from the published reviewer reports and rebuttal letters prior to publication. If your manuscript has been previously reviewed at another journal, those Reviewers' comments would not form part of the published peer review file.

*

REVIEWERS' COMMENTS:

Reviewer #2 (Remarks to the Author):

Thank for a thorough revision and congratulations on a great paper.

Reviewer #3 (Remarks to the Author):

I am pleased to report that the authors did a good job of addressing my comments. Just a couple very minor points as they finalize this paper:

1. Although they tempered down the explicit causal language even further in reporting the results, the authors clearly think there is a causal relationship between zero-sum thinking and avoidance of political conversation. Explicit claims are carefully guarded, but the causal notion is in the background of the paper, including the introduction (which explains several reasons why they believe zero-sum thinking would lead to fewer political conversations), results (which include a mediation analysis attempting to "explain" the relationship between these variables using a particular causal chain), and discussion (which suggests that intervening on zero-sum beliefs would be a good way to address the problem of political polarization). I think all of this is absolutely fine. But it creates a bit of a mismatch. My preferred method, rather than using ever more cautious language in the results section (like "statistically predicts") would be to offer a potential causal interpretation in the discussion section, along with the authors' reasons for thinking the relationship is mostly likely to be causal, along with the reasons why confidence in this conclusion is necessarily

limited at this time. To me, that would do a better job of communicating the authors' interpretation clearly without making stronger definite claims than are merited by the data. All this said, I leave this up to the authors and editor and I am okay signing off on this aspect of the paper as-is.

2. A couple (hopefully!) very small statistics points I noticed on this round due to changes made to the manuscript:

(i) The low end on two of the confidence intervals on page 8 are reported as '0' with no decimal points. At the very least the authors need to report the appropriate number of significant figures, but I also suspect these aren't quite right because the p-values in both cases are close-ish but not super-close to .05 (.039 and .052; so the first lower bound should be a little more than 0 and the second a little less than 0).

(ii) I don't quite understand how the regression adjusted for political party, since it looks like this is treated as a continuous variable in the supplementary regression tables, but the methods section state that this was measured as a categorical variable with 13 levels. I doubt the authors would have inadvertently treated this as a continuous variable (!) but some clarity is needed to know exactly what they did. In the previous round I had assumed this was measured continuously, but evidently not according to the new edits.

Thanks for the opportunity to read your paper – looking forward to seeing it in print and I hope it can help make the world talk a little more!

Response to Reviews

We are very thankful for the reviews by the two anonymous reviewers in this final round of revisions. We are especially grateful for Reviewer 2's comment, which congratulated us on a "*great paper*", and for both reviewers' belief that we "*did a good job*" of addressing their comments in a "*thorough revision.*" We also join Reviewer 3 in their "*hope [that] it can help make the world talk a little more!*"

Of course, Reviewer 3 raised a couple of helpful and constructive comments that strengthen the manuscript. Below, we outline these comments and describe in detail how we addressed them. As a result, we believe that the manuscript is substantially stronger and more impactful.

Reviewer 2

"Thank for a thorough revision and congratulations on a great paper."

Thank *you* very much for your time and thoughtful feedback on the paper. Your detailed and insightful reviews throughout this process made our paper stronger and your suggestions and advice helped us to enhance the paper's theoretical and empirical foundations.

Reviewer 3

"I am pleased to report that the authors did a good job of addressing my comments."

We wholeheartedly thank Review 3 for this feedback! We are extremely happy to hear that the reviewer found the revisions relevant and impactful to the paper.

“Although they tempered down the explicit causal language even further in reporting the results, the authors clearly think there is a causal relationship between zero-sum thinking and avoidance of political conversation. Explicit claims are carefully guarded, but the causal notion is in the background of the paper, including the introduction (which explains several reasons why they believe zero-sum thinking would lead to fewer political conversations), results (which include a mediation analysis attempting to “explain” the relationship between these variables using a particular causal chain), and discussion (which suggests that intervening on zero-sum beliefs would be a good way to address the problem of political polarization). I think all of this is absolutely fine. But it creates a bit of a mismatch. My preferred method, rather than using ever more cautious language in the results section (like “statistically predicts”) would be to offer a potential causal interpretation in the discussion section, along with the authors’ reasons for thinking the relationship is mostly likely to be causal, along with the reasons why confidence in this conclusion is necessarily limited at this time. To me, that would do a better job of communicating the authors’ interpretation clearly without making stronger definite claims than are merited by the data. All this said, I leave this up to the authors and editor and I am okay signing off on this aspect of the paper as-is.”

We greatly appreciated Reviewer 3’s comment on this issue. We agree that it is important to communicate the results without making definitively strong claims about causality, and in

revising the manuscript we have done exactly that. Specifically, as suggested by Reviewer 3, we added a paragraph to the Discussion in which we explicitly note the limitations of the reported studies and the importance of further research on the topic. This addition can be found on page 14 of the revised manuscript.

“A couple (hopefully!) very small statistics points I noticed on this round due to changes made to the manuscript:

(i) The low end on two of the confidence intervals on page 8 are reported as ‘0’ with no decimal points. At the very least the authors need to report the appropriate number of significant figures, but I also suspect these aren’t quite right because the p-values in both cases are close-ish but not super-close to .05 (.039 and .052; so the first lower bound should be a little more than 0 and the second a little less than 0).”

Thank you for raising this concern. In revising the manuscript, we made sure to fix this issue and report the appropriate number of decimal points in all confidence intervals (p. 8). In addition, we went through the entire paper to make sure that all other confidence intervals are fully reported.

“(ii) I don’t quite understand how the regression adjusted for political party, since it looks like this is treated as a continuous variable in the supplementary regression tables, but the methods section state that this was measured as a categorical variable with 13 levels. I doubt the authors would have inadvertently treated this as a continuous variable (!) but

some clarity is needed to know exactly what they did. In the previous round I had assumed this was measured continuously, but evidently not according to the new edits.”

Thank you for bringing this to our attention! Indeed, for some reason our analyses treated political affiliation as a continuous rather than a categorical variable, as it should have done. In revising the manuscript, we made sure to correct this issue, making it clear that political party identification and political affiliation are treated as categorical variables. To address this, we have revised all the statistical tables in the Supplementary Materials and, where relevant, all the reported statistics in the main text. We apologize for any confusion caused by this error and are extremely grateful that we were able to correct it! Thank you once again for your attention to detail!

“Thanks for the opportunity to read your paper – looking forward to seeing it in print and I hope it can help make the world talk a little more!”

We wholeheartedly share your hope, which has been the impetus for this project from day one.

Thank you!

13th Mar 24

Dear Dr Davidai,

Your manuscript titled "Zero-Sum Beliefs and the Avoidance of Political Conversations" has now been seen by Reviewer #3, whose comments appear below. In light of their advice I am delighted to say that we are happy, in principle, to publish a suitably revised version in Communications Psychology under the open access CC BY license (Creative Commons Attribution v4.0 International License).

We therefore invite you to revise your paper one last time to address the remaining concerns of our reviewers and a list of editorial requests. At the same time we ask that you edit your manuscript to comply with our format requirements and to maximise the accessibility and therefore the impact of your work.

We ask in particular for a revision of the presentation of the statistics to address the confusion about what analyses support which interpretation (and how measures are derived) identified by Reviewer #3. This entails that we strongly recommend that you adopt the standard Article format for Communications Psychology [Abstract - Introduction - Methods - Results - Discussion], include much more detail on the analyses in the Methods (which presently focus on the data acquisition and characteristics), and include the critical statistics that are currently in the SI in the main manuscript. There is no word limit applied to the Methods section and the number of permissible display items is 10 (Figures and Tables combined). In general terms, any finding that is interpreted should be supported by statistics in the main manuscript, not the SI. More detailed recommendations are included in the attached file.

EDITORIAL REQUESTS:

Please review our specific editorial comments and requests regarding your manuscript in the attached "Editorial Requests Table". Please outline your response to each request in the right-hand column. Please upload the completed table with your manuscript files as a Related Manuscript file.

SUBMISSION INFORMATION:

OPEN ACCESS:

Communications Psychology is a fully open access journal. Articles are made freely accessible on publication under a [CC BY license](https://creativecommons.org/licenses/by/4.0/) (Creative Commons Attribution 4.0 International License). This license allows maximum dissemination and re-use of open access materials and is preferred by many

research funding bodies.

For further information about article processing charges, open access funding, and advice and support from Nature Research, please visit <https://www.nature.com/commspsychol/article-processing-charges>

At acceptance, you will be provided with instructions for completing this CC BY license on behalf of all authors. This grants us the necessary permissions to publish your paper. Additionally, you will be asked to declare that all required third party permissions have been obtained, and to provide billing information in order to pay the article-processing charge (APC).

* TRANSPARENT PEER REVIEW: Communications Psychology uses a transparent peer review system. On author request, confidential information and data can be removed from the published reviewer reports and rebuttal letters prior to publication. If you are concerned about the release of confidential data, please let us know specifically what information you would like to have removed. Please note that we cannot incorporate redactions for any other reasons.

* CODE AVAILABILITY: All Communications Psychology manuscripts must include a section titled "Code Availability" at the end of the methods section. We require that the custom analysis code supporting your conclusions is made available in a publicly accessible repository at this stage; please choose a repository that generates a digital object identifier (DOI) for the code; the link to the repository and the DOI must be included in the Code Availability statement. Publication as Supplementary Information will not suffice.

* DATA AVAILABILITY:

[link redacted]

Best regards,

Marika

Marike Schiffer, PhD
Chief Editor
Communications Psychology

REVIEWERS' COMMENTS:

Reviewer #3 (Remarks to the Author):

Thanks for addressing my previous comments. I am genuinely sorry for harping on this, but I am still a bit confused about political orientation, and I do think this is a potentially important confound that readers will want to be assured is not driving the results.

Looking through the supplementary materials, I see that political affiliation is treated as a many-level categorical variable in Tables S1 and S2 (by using 12 dummy-coded variables), as a 3-level categorical variable in Tables S3, S5, S8, and S10, and as one "political orientation" variable in Tables S4, S6, S9, and S11. Looking at the methods, I see that this measured as a list of 13 options in Study 1, and measured in two different ways in Study 2. Could the authors please write the methods for Study 2 more clearly so that it is clear what the response options are?

* Presumably the first political affiliation measure is continuous (since it was used to calculate political extremism)? What is the scale and what are the anchors?

* I gather that the second political affiliation measure is categorical... But what are the categories? It sounds like it would be "Strong Republican/Democrat" or "Not a strong Republican/Democrat," but in the analyses the dummy-coded variable is called "Republican," which makes me think there are three options (Democrat, Republican, and Independent)... but then why is the question phrased that way?

* One of these seems to be called "political orientation" and the other "political affiliation" in the main text, but they are both called "political affiliation" in the methods and supplement. I think the continuous one is called "orientation" and the categorical one "affiliation," but I still think this is confusing and I would appreciate if this terminology could be more transparent and consistent.

Finally, looking through the methods in more detail, I also see that the participants seem to have self-reported their political orientation at both the beginning and end of the study (assuming the procedure is written in chronological order). Was that just an error in the coding of the study, or was there an additional measure of political orientation measured at the same time as age, gender, etc., which was not analyzed?

Once again, sorry for zooming in on something that probably isn't all that important, but I do think this is confusing enough and of sufficient interest to readers that we need to be crystal clear on the procedure and analyses for political beliefs.

Response to Reviews

Editors' comments:

Response: The Editor made several comments across the manuscript, which we made sure to address during this revision process. First, as requested by the Editor, we expanded the Methods sections of both Studies 1 and 2 in order to give the readers more information regarding the measures used and the analyses conducted. Second, as suggested, we moved all the relevant tables from the Supplementary Materials to the manuscript, which we believe will help readers better interpret the findings described in the paper. Third, we added 'Data Availability,' 'Code Availability,' 'Author Contributions,' 'Competing Interests,' and 'Acknowledgements' sections to the manuscript. Finally, we made all the edits and additions in the main text, as suggested by the Editor.

Reviewer #3 (Remarks to the Author):

Thanks for addressing my previous comments. I am genuinely sorry for harping on this, but I am still a bit confused about political orientation, and I do think this is a potentially important confound that readers will want to be assured is not driving the results.

Looking through the supplementary materials, I see that political affiliation is treated as a many-level categorical variable in Tables S1 and S2 (by using 12 dummy-coded variables), as a 3-level categorical variable in Tables S3, S5, S8, and S10, and as one "political orientation" variable in Tables S4, S6, S9, and S11. Looking at the methods, I see that this measured as a list of 13 options in Study 1, and measured in two different ways in Study 2. Could the authors please write the methods for Study 2 more clearly so that it is clear what the response options are?

Response: We are always happy to increase the clarity of the manuscript, and in revising the manuscript have followed this advice and done exactly that. Specifically, we revised the description for Study 2 to explicitly clarify what each variable measured. These edits can be found on pages 9-10 of the revised manuscript. In addition, while revising the manuscript we also added more details to the description of the political variable measure in Study 1, which can be found on page 8 of the revised manuscript.

** Presumably the first political affiliation measure is continuous (since it was used to calculate political extremism)? What is the scale and what are the anchors?*

** I gather that the second political affiliation measure is categorical... But what are the categories? It sounds like it would be "Strong Republican/Democrat" or "Not a strong Republican/Democrat," but in the analyses the dummy-coded variable is called "Republican," which makes me think there are three options (Democrat, Republican, and Independent)... but then why is the question phrased that way?*

** One of these seems to be called "political orientation" and the other "political affiliation" in the main text, but they are both called "political affiliation" in the methods and supplement. I think the continuous one is called "orientation" and the categorical*

one “affiliation,” but I still think this is confusing and I would appreciate if this terminology could be more transparent and consistent.

Response: We appreciate the opportunity to clarify this issue. As described above, we elaborated on the description of the methods for both studies reported in the manuscript, in order to give readers more information about these measures. As we now note in the revised manuscript, participants in Study 1 reported their political party affiliation on a categorical variable, identifying the political party that they most closely support among the 13 different parties that participated in the 2022 elections to Israel’s parliament (the “Knesset”; “Generally speaking, which of the following political parties do you support? ”: Ha’Likkud, Ha’Avoda, Yesh Atid, HaZionot Ha’Datit, Meretz, Ra’am, Ha’Machane Ha’Mamlachti, Hadash-Ta’al, Shas, Ha’Bait Ha’Yehudi, Yahadot Ha’Torah, Balad, Israel Beitenu, and other). In Study 2, participants completed two measures. First, participants indicated, on a three-level categorical variable, their political party affiliation by reporting the political party with which they most closely identify in the U.S. political map (“Generally speaking, how do you usually think of yourself in terms of political affiliation? ”; Republican, Democrat, or Independent). Second, participants indicated the strength of their political affiliation (for Republicans and Democrats; “Would you call yourself a strong Republican/Democrat, or not a very strong Republican/Democrat? ”) or their general political leanings (for independents; “If you had to choose, would you say that you lean more towards Republicans or Democrats? ”).

Finally, looking through the methods in more detail, I also see that the participants seem to have self-reported their political orientation at both the beginning and end of the study (assuming the procedure is written in chronological order). Was that just an error in the coding of the study, or was there an additional measure of political orientation measured at the same time as age, gender, etc., which was not analyzed?

Response: The additional measure that Reviewer 3 is referring to is not a measure of political orientation or affiliation but rather a general question of whether participants planned to vote in the coming Israeli or U.S. elections, respectively (*yes, no, or I have yet to decide*). Participants in Study 2 also indicated at this point their political ideology on a 7-point Likert scale (ranging from very liberal to very conservative), which led to very similar results as the categorical political party affiliation measure.

Once again, sorry for zooming in on something that probably isn’t all that important, but I do think this is confusing enough and of sufficient interest to readers that we need to be crystal clear on the procedure and analyses for political beliefs.

Response: Thank you so much for your attention to details, which we believe have helped make our paper stronger!